# SEPARABLE POLICY LEARNING FOR EMERGENCY VEHICLE PRIORITIZED TRAFFIC SIGNAL CONTROL

## ABSTRACT

Traffic Signal Control plays a vital role in optimizing urban traffic flow and reducing accidents by regulating signal phases at intersections. While traditional fixed-time control methods are simple and infrastructure-efficient, they fail to adapt to complex and dynamic traffic patterns, particularly during peak periods or in the presence of emergency vehicles. In this paper, we address the emergency-vehicle-aware traffic signal control problem by proposing a decoupled policy fusion framework that separately optimizes control strategies for regular vehicles and emergency vehicles. The two policies are later combined into a global strategy with automatically learned weights, mitigating the negative impact of $Q$-function approximation errors. We further introduce **SplitEMV**, a novel multi-agent model that enhances inter-agent communication and decision efficiency. Experiments demonstrate that our method significantly improves emergency vehicle response times while preserving efficiency of regular vehicles. The learned emergency vehicle prioritized policy also integrates seamlessly with existing traffic signal control methods in a zero-shot manner, supporting practical deployment.

## 1 INTRODUCTION

Traffic signal control (TSC) plays a crucial role in optimizing traffic flow and enhancing road safety within modern urban settings. By coordinating signal phases at congested intersections, TSC alleviates congestion resulting from conflicting traffic streams Eom & Kim (2020), facilitates orderly traffic movement Wang et al. (2018), and reduces the risk of collisions Du et al. (2023). Over the years, numerous approaches have been put forward to tackle the TSC problem. Traditional TSC methods Albatish & Abu-Naser (2019); Majstorović et al. (2023), recognized for their simplicity and minimal infrastructure demands, have been widely implemented. As urban traffic grows increasingly intricate and vehicle volumes surge, these methods prove inadequate in adapting to real-time traffic dynamics. Although adaptive TSC methods Wang et al. (2018); Cools et al. (2013) utilize modern sensing technologies to adjust signal timings based on real-time traffic states, their dependence on domain knowledge from experts limits the effectiveness compared to flexible learning-based methods.

As the TSC problem can be formalized as Markov Decision Process, solving TSC problems with Reinforcement Learning (RL) becomes a popular way. Classical RL-based methods El-Tantawy et al. (2013); Chin et al. (2012); Araghi et al. (2013) have achieved promising outcomes in adaptive TSC. However, they encounter scalability challenges owing to large state spaces and high sample complexity. Recently, deep RL has outperformed traditional TSC methods in dynamic traffic scenarios Yau et al. (2017); Wei et al. (2021); Li et al. (2021); Zhao et al. (2024); Zhang et al. (2024). These methods take the full state as input and utilize deep neural networks to approximate cumulative rewards or policies. This enables the methods to control traffic signals according to real-time traffic conditions. Nevertheless, their reliance on manually tuned parameters frequently restricts their efficacy.

Although existing studies have significantly enhanced the efficiency of regular vehicles (RVs), real-world scenarios involve the priority passage of emergency vehicles (EMVs), which need to their destinations in the shortest possible time. EMVs are permitted to violate traffic rules in order to reduce the travel time to their destinations. However, during congestion, they may be impeded by regular traffic, diminishing their advantage and increasing the risk of accidents. Consequently, optimizing traffic signals to assist EMVs in traversing intersections is of utmost importance.

Numerous studies have concentrated on enhancing the efficiency of EMVs through diverse approaches, including route planning Muzzini & Montangero (2024); Peelam et al. (2024) and signal preemption strategies Lu & Wang (2019); Humagain et al. (2020). However, these studies do not integrate adaptive TSC, which might result in sub-optimal policies. Existing regular TSC methods solely focus on improving the efficiency of RVs and fail to enhance the efficiency of EMVs. The state-of-the-art method, EMVLight Su et al. (2022; 2023), combines real-time EMV routing with adaptive TSC. Nevertheless, its effectiveness is limited by complex multi-level signal architectures.

To address the aforementioned issues, we propose **Decoupled Learning and Adaptive Strategy Merging**. This approach enables the flexible integration of strategies prioritizing RVs and EMVs. It significantly enhances the efficiency of EMVs while having a minimal impact on RVs. We implement Decoupled Learning to separately learn strategies for RV and EMV objectives. Subsequently, we utilize **Adaptive Strategy Merging** to smoothly integrate these two strategies. Our fusion strategy is fully automated and does not depend on manual parameter tuning. Beyond TSC, our method is also applicable to other multi-objective problems. Particularly when the objectives are independent, it facilitates efficient task decomposition and improves the model's ability to achieve multiple goals.

The key contributions of our method are summarized as follows: (1) We propose Decoupled Learning and Adaptive Strategy Merging strategy for TSC problem. This strategy significantly enhances the efficiency of EMVs while having a minimal impact on RVs. (2) Based on this framework, we develop SplitEMV, a multi-stage trained TSC method. It incorporates a zero-shot merging technique to seamlessly integrate EMV-prioritized strategies with existing RV-prioritized methods. (3) Extensive experiments on public benchmarks show that our method achieves state-of-the-art result, significantly improving traffic efficiency for both RVs and EMVs and showcases the ability to integrate EMV-prioritized capabilities for all existing methods in a zero-shot manner.

## 2    PRELIMINARIES

We provide a formal definition of the EMV-aware TSC problem in this section, which extends the conventional multi-intersection TSC problem. We begin by introducing Decentralized Partially Observable Markov Decision Process (Dec-POMDP), then briefly describe the standard TSC formulation and introduce its extension to incorporate EMVs. We model the TSC task as a Dec-POMDP, the detailed definition of Dec-POMDP can be found in Appendix B.

The objective of a TSC problem is to minimize the average travel time of vehicles in a road network, and its formal definition is as follows:

**Problem 2.1** (Traffic Signal Control Problem). *The environment consists of intersections $\mathcal{I}$, roads $\mathcal{R}$, and vehicles $\mathcal{V}$. Each intersection $I_i \in \mathcal{I}$ is controlled by an agent $A_i$, which executes an action every $\Delta t$ time steps based on its policy $\pi_i$. At time $t$, agent $A_i$ receives a partial observation $z_i^t$ and selects an action $a_i \in \mathcal{A}_i$ to determine the next signal phase.*

Definitions of *Intersection*, *Road*, *Traffic Signal Phase*, and *Vehicle* in the context of the TSC problem are provided in Appendix C. We now extend the above problem to incorporate EMVs, which are a subset of $\mathcal{V}$ that require prioritized treatment. The definition of *Emergency Vehicle* and the EMV-Aware TSC problem is given as follows:

**Definition 2.1** (Emergency Vehicle). *An **emergency vehicle** $E_i \in \mathcal{E} \subseteq \mathcal{V}$ is a special vehicle selected manually or randomly. Its behavior is identical to that of regular vehicles, but in the EMV-aware TSC problem, its travel time is explicitly prioritized for minimization.*

**Problem 2.2** (EMV-Aware Traffic Signal Control Problem). *The environment is composed of intersections $\mathcal{I}$, roads $\mathcal{R}$, vehicles $\mathcal{V}$, and emergency vehicles $\mathcal{E}$. Each intersection $I_i \in \mathcal{I}$ is controlled by an agent $A_i$, which executes actions every $\Delta t$ seconds based on policy $\pi_i$. At time $t$, agent $A_i$ observes partial state $z_i^t$ and selects an action $a_i \in \mathcal{A}_i$ to determine the next phase.*

For the EMV-Aware TSC problem, our objective is to minimize the average travel time of EMVs, while keeping the impact on the average travel time of RVs to a minimum. We formulate this problem as a Dec-POMDP, with the detailed definitions of the state space $s$, observation $z$, action space $a$, and joint reward function $r$ provided in Appendix B. Since the goal is to reduce the average travel time of both RVs and EMVs, the problem can be naturally framed as a multi-objective reinforcement

learning problem, and the reward function $r_i^t$ for $i$-th intersection at timestamp $t$ can be defined as:

$$r_i^t = r_i^{n,t} + \beta r_i^{e,t} \tag{1}$$

where $r_i^{n,t}$ and $r_i^{e,t}$ are rewards of RVs and EMVs, and $\beta$ controls the weight of EMV priority.

## 3 METHODOLOGY

Since the strategy needs to optimize EMVs while minimizing the impact on RVs, based on Equation 1, the optimal strategy for the EMV-aware TSC problem should also be optimal for the regular TSC problem, as the reward function remains identical in the absence of EMVs. However, with the introduction of EMVs, the weight $\beta$ influences the performance of the learned strategy. A large $\beta$ causes the model to prioritize EMVs, and potentially degrading the performance for RVs. Conversely, a small $\beta$ leads the model to ignore EMVs. To address this issue, we first decouple the control strategies for RVs and EMVs in Sec. 3.1 and then propose a merging method to integrate the two strategies effectively in Sec. 3.2. Finally, we propose the SplitEMV model and a multi-stage training strategy to learn the decoupled strategies.

### 3.1 DECOUPLED LEARNING FOR ROBUST EMV-AWARE CONTROL

Training a model with the joint reward function that simultaneously considers RVs and EMVs leads to unstable performance. To mitigate this, we decouple the control strategies, enabling the model to better distinguish and learn each task. The key difference between the regular and EMV-aware TSC problems lies in the reward function: the latter introduces an additional weighted emergency vehicle reward. To ensure that model performance remains independent of the weight $\beta$, it is essential to separate the two reward components. Therefore, we obtain the optimal strategy $\pi^*$ of the EMV-aware TSC problem by substituting the cumulative reward function in reinforcement learning $\pi^* = \arg\max_\pi \mathbb{E}\left(\sum_{t=0}^\infty \gamma^t r^t\right)$ with the following reformulated reward:

$$\pi^* = \arg\max_\pi \mathbb{E}\left(\sum_{t=0}^\infty \gamma^t \left(r^{n,t} + \beta r^{e,t}\right)\right) = \arg\max_\pi \left(R_N + \beta R_E\right) \tag{2}$$

where $r^t$ is the reward at time $t$, and $\gamma$ is the discount factor. $r^{n,t}$ and $r^{e,t}$ are the rewards of RVs and EMVs at time $t$. We also define $R_N = \mathbb{E}\left(\sum_{t=0}^\infty \gamma^t r^{n,t}\right)$ and $R_E = \mathbb{E}\left(\sum_{t=0}^\infty \gamma^t r^{e,t}\right)$ as the cumulative rewards of RVs and EMVs, respectively.

We decouple the two tasks based on the cumulative reward function from value-based reinforcement learning methods, which compute the state-action value function ($Q$-function) and select actions accordingly. The model is trained to minimize the loss of $Q$-function based on the Bellman equation. The loss function $\mathcal{L}_Q = \mathrm{MSE}\left(Q(s,a), r^n + \beta r^e + \gamma \max_{a'} Q(s',a')\right)$, and $Q$-function $Q^*(s,a)$ for the optimal strategy can be expressed as:

$$Q^*(s,a) = \arg\min_{Q(s,a)} \mathbb{E}\left[\left((Q(s,a) - \gamma Q(s',a')) - (r^n + \beta r^e)\right)^2\right] = \mathbb{E}\left[R_N^Q + \beta R_E^Q\right] \tag{3}$$

where $s$ and $a$ represent the current state and action, respectively, $s'$ and $a'$ is the next state and next action, $\beta$ is the weight assigned to the emergency vehicle reward, and $\gamma$ is the discount factor.

To improve the performance of EMVs while maintaining regular vehicle performance, we first consider the case without EMVs, i.e., $r^e = 0$. In this case, the loss function $\mathcal{L}_N = \mathrm{MSE}\left(Q_N(s,a), r^n + \gamma \max_{a'} Q_N(s',a')\right)$ and the optimal $Q$-function $Q_N^*(s,a)$ for RVs are:

$$Q_N^*(s,a) = \arg\min_{Q_N(s,a)} \mathbb{E}\left[\left((Q_N(s,a) - \gamma Q_N(s',a')) - r^n\right)^2\right] = \mathbb{E}\left[R_N^N\right]. \tag{4}$$

Based on Eq. 3 and Eq. 4, we can also define an EMV prioritized policy, the loss function $\mathcal{L}_E = \mathrm{MSE}\left(Q_E(s,a), r^e + \gamma \max_{a'} Q_E(s',a')\right)$ and the optimal $Q$-function $Q_E^*(s,a)$ for EMVs are:

$$Q_E^*(s,a) = \arg\min_{Q_E(s,a)} \mathbb{E}\left[\left((Q_E(s,a) - \gamma Q_E(s',a')) - r^e\right)^2\right] = \mathbb{E}\left[R_E^E\right]. \tag{5}$$

$R_X^Y$ in Eqs. 3, 4, and 5 denotes the cumulative reward of type $X$ when using $Y$ as the action policy. To integrate the two tasks, we define the combined $Q$-function as:

$$Q(s,a) = Q_N(s,a) + \beta Q_E(s,a), \tag{6}$$

where $Q_N$ and $Q_E$ are trained independently using $\mathcal{L}_N$ and $\mathcal{L}_E$, respectively. $Q(s,a)$ is equivalent to $Q^*(s,a)$, under certain conditions, and the detailed discussion is in Appendix H.

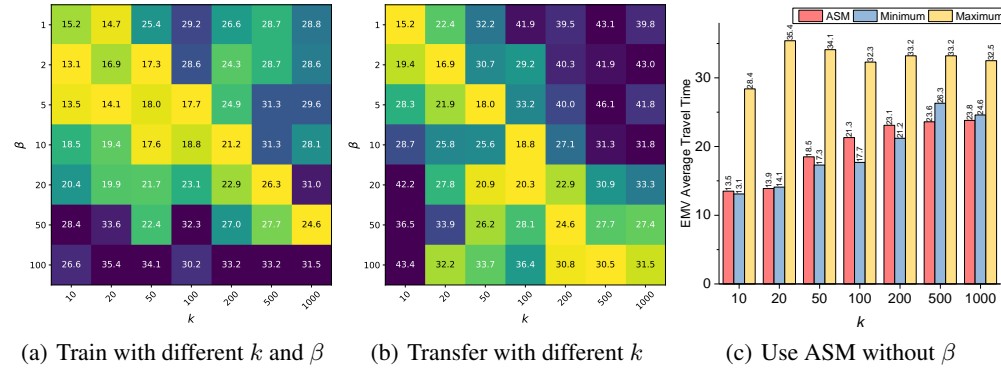

(a) Train with different $k$ and $\beta$    (b) Transfer with different $k$    (c) Use ASM without $\beta$

Figure 1: Average travel time for EMVs in a single-intersection environment; lower is better. (a) Model is trained using different initial vehicle numbers $k$ and EMV priority weights $\beta$. (b) Model is trained with different $k$ and $\beta = 0.1k$, then directly transferred to environments with varying $k$. (c) Use ASM to adaptively merge $Q_N$ and $Q_E$, results are compared with the min/max average travel times for the same $k$ but different $\beta$. ASM performs closely to the minimum result.

## 3.2 ADAPTIVE STRATEGY MERGING FOR ROBUST EMV-AWARE CONTROL

The selection of the weight $\beta$ in Eq.6 has a substantial influence on the overall performance. In this section, we conduct a detailed analysis of the impact of $\beta$ and present Adaptive Strategy Merging (ASM). Specifically, our objective is to integrate the existing $Q_N(s, a)$ and $Q_E(s, a)$ into a unified $Q(s, a)$ without depending on a fixed, pre-defined $\beta$. For the sake of simplicity, we omit the explicit notation of $(s, a)$ when its context is evident.

We first present the following two propositions, and their proofs can be found in Appendix G.

**Proposition 3.1.** *Assume $Q = Q_1 + Q_2$, where $Q_1$ and $Q_2$ are two different Q functions,* $\mathrm{second\_max}(Q)$ *is the second-largest value in Q, When Eq. 7 holds, the position of the element with the maximum value in Q is the same as that in $Q_1$.*

$$\max(Q_1) - \mathrm{second\_max}(Q_1) > \max(Q_2) - \min(Q_2) \tag{7}$$

**Proposition 3.2.** *Given Q and any $\mu, \sigma^2$, where $\mu \in \mathbb{R}$, $\sigma^2 \in \mathbb{R}^+$. Define $Q' = \frac{Q - \mu}{\sigma^2}$, then the strategy based on $Q'$ is identical to the strategy based on Q.*

A good strategy for the EMV-aware TSC problem should exhibit the same performance as the regular TSC problem in the absence of EMVs, and the strategy should give precedence to EMVs when EMVs are present. Formally, when there is no influence from EMVs, we have $\arg\max_a Q = \arg\max_a Q_N$, and when the strategy is influenced by EMVs, $\arg\max_a Q = \arg\max_a Q_E$.

To achieve this for $Q$, we must ensure that Proposition 3.1 holds. Subsequently, we can guarantee that $Q$ accurately selects the action corresponding to the maximum value of $Q_1$, and the addition of $Q_2$ does not alter the maximum-valued action of $Q_1$. It is worth noting that this is a sufficient condition. When $\max(Q_2) > \max(Q_1) - \mathrm{second\_max}(Q_1)$, since the action associated with $\max(Q_2)$ may differ from that of $\mathrm{second\_max}(Q_1)$, the maximum value of $Q$ might still be the same as that of $Q_1$.

To enhance the probability that Proposition 3.1 holds, we normalize $Q_N$ and $Q_E$ in accordance with the principle stated in Proposition 3.2. This normalization is of vital importance when integrating $Q_N$ and $Q_E$ to form $Q$. In the equation $Q = Q_N + \beta Q_E$, a fixed $\beta$ fails to satisfy Eq. 7 under different traffic densities. This is because neural network approximation errors affect $Q_N$ and $Q_E$ differently, which is involved by the average number of vehicles $k$. As a result, a fixed $\beta$ causes performance degradation outside a narrow range of $k$. Fig. 1 illustrates this phenomenon in a simplified single-intersection environment. In (a), models trained with different $k$ and $\beta$ indicate that the optimal $\beta$ scales with $k$, and extreme $\beta$ values deteriorate performance. In (b), models trained with the optimal $\beta$ exhibit poor generalization across unseen $k$. In (c), a model trained at $k = 100$ and augmented by our proposed ASM maintains low EMV travel times close to the best $\beta$ selection across various $k$. The details of the setup and evaluation are presented in Appendix D.

Figure 2: SplitEMV model structure, training strategy and adaptive strategy merging.

Based on the foregoing analysis, we propose the ASM, which normalizes $Q_N$ and $Q_E$ according to their data-distribution characteristics and then merges them to obtain $Q$. Regarding $Q_N$, when the number of vehicles varies, the mean and variance of $Q_N$ are positively correlated with the number of vehicles. Therefore, when $k$ is large, the impact of $Q_E$ on the strategy is negligible. Conversely, when $k$ is small, the variance of $Q_E$ may significantly disrupt the optimal-strategy selection of $Q$. Consequently, we utilize the mean and variance of all actions of $Q_N$ in the current state for normalization as Eq. 8. Through this normalization, the mean and variance of $Q'_N$ are 0 and 1 respectively, effectively eliminating the influence of $k$ on $Q_N$.

$$Q'_N(s,a) = \frac{Q_N(s,a) - \frac{1}{|A|} \sum_{a \in A} Q_N(s,a)}{\sqrt{\frac{1}{|A|} \sum_{a \in A} \left(Q_N(s,a) - \mu_N\right)^2}} \qquad (8)$$

For $Q_E$, if we directly normalize it with respect to its actions, the variance will be amplified, because when there is no EMV, $Q_E$ is close to 0, and normalization will magnify the noise. On the other hand, when there are EMVs, the anticipated action will have a relatively larger value compared to other actions, and normalization will reduce its impact. Therefore, rather than normalizing $Q_E$ based on its actions, we first conduct a one-epoch simulation in the environment with the default $\beta$ to calculate the mean and variance of $Q_E$. The default value of $\beta$ is set to 1. Additional experiments presented in Appendix F demonstrate that varying $\beta$ has a negligible effect on the final strategy. We define $\overline{\sigma_E}$ as the average of the $m$ largest variances of $Q_E$, where $m$ is the number of states containing EMVs. The normalization of $Q_E$ is carried out as shown in Eq. 9.

$$Q'_E(s,a) = \frac{Q_E(s,a) - \frac{1}{|A|} \sum_{a \in A} Q_E(s,a)}{\overline{\sigma_E}} \qquad (9)$$

Then, we can directly calculate $Q(s,a) = Q_N(s,a) + Q_E(s,a)$. In Appendix E, we present an analysis of the values of $Q_N$ and $Q_E$ before and after normalization. With Decoupled Learning and ASM, the model can select the optimal action and reduce the interference of the training error. This method can operate without any hyper-parameters and is generally applicable to different scenarios. In Appendix F, we show that during one-epoch simulation, $\overline{\sigma_E}$ is a statistic of the EMV model and is largely independent of testing scenarios. This indicates that evaluations can be conducted on simulated scenarios and applied directly to real-world situations.

### 3.3 SplitEMV Model

Based on the foregoing analysis, we propose a model that is applicable to both $Q_N$ and $Q_E$. A more elaborate version is presented in Appendix J. The overall structure, depicted in the left of Fig. 2, comprises two primary components: communication information generation and $Q$-value estimation.

The first component decomposes the input state into individual incoming lanes. Each lane encompasses vehicle count, direction, signal phase, and, if applicable, EMV information. An MLP is utilized to encode each lane into $h_i^L$. Subsequently, a multi-head attention is employed to capture the interactions among lanes. To further strengthen communication, the model predicts the contribution of each lane to adjacent intersections and categorizes them according to their outgoing directions.

The second component combines the communication information received from neighboring agents with the current local state to predict $Q$-values. It includes external information in addition to what

the first component has. For each action $a_i$, the lane vectors are grouped according to whether the corresponding lanes are allowed under the current signal phase. The vectors within each group are averaged, and then concatenated with an embedding that indicates the activation status of $a_i$. An MLP is subsequently utilized to generate the $Q$-value prediction.

There are differences when applying the model to $Q_N$ and $Q_E$. For $Q_N$, the EMV-related inputs are set to zero. This enables the model to concentrate solely on RV optimization. For $Q_E$, the EMV inputs are set to their actual observed values.

## 3.4 TRAINING STRATEGY

To train the model, A straightforward approach is to train the model using the combined reward of RVs and EMVs. The loss function $\mathcal{L}_{\text{combine}}$ is as follows:

$$\mathcal{L}_{\text{combine}} = \text{MSE}\left(Q_N(s,a) + Q_E(s,a), (\boldsymbol{r}^n + \boldsymbol{r}^e) + \gamma \max_{a'} \left(Q_N(s',a') + Q_E(s',a')\right)\right) \quad (10)$$

However, using $\mathcal{L}_{\text{combine}}$ will result in $Q_N$ and $Q_E$ being unable to distinguish their respective rewards, and thus unable to decouple them. An alternative is to ensure the decoupling using $\mathcal{L}_{\text{split}}$.

$$\mathcal{L}_{\text{split}} = \text{MSE}\left(Q_N(s,a), \boldsymbol{r}^n + \gamma \max_{a'} Q_N(s',a')\right) + \text{MSE}\left(Q_E(s,a), \boldsymbol{r}^e + \gamma \max_{a'} Q_E(s',a')\right) \quad (11)$$

$\mathcal{L}_{\text{split}}$ supervises the training of $Q_N$ and $Q_E$ with their respective rewards. However, in practice, $\mathcal{L}_{\text{split}}$ does not perform well for the equation implicitly assumes $\beta = 1$. From analysis in 3.2, setting a constant weight will result in a sub-optimal strategy when the average number of vehicles at the intersection varies, reducing the learning efficiency and stability of the $Q$-function, especially the RV model. Through experiments, we can observe a significant performance gap using $\mathcal{L}_{\text{split}}$. Therefore, we propose a multi-stage training strategy, which consists of RV model learning stage, EMV model initialization stage, and joint training stage, as depicted in the middle of Fig. 2.

In RV model learning stage, we solely train $Q_N$, which is focused on RVs only and is identical to other TSC methods. Its loss function $\mathcal{L}_{\text{RV}}$ is given by Eq. 12, where EpsRand represents an $\epsilon$-greedy strategy, which will substitute the selected action by a random action with a probability of $\epsilon$.

$$\mathcal{L}_{\text{RV}} = \text{MSE}\left(Q_N\left(s, \text{EpsRand}\left(\arg\max_{a'} Q_N(s,a'), \epsilon\right)\right), \boldsymbol{r}^n + \gamma \max_{a'} Q_N(s',a')\right) \quad (12)$$

In EMV model initialization stage, we train $Q_E$ to learn the strategy for EMVs. Since the occurrence frequency of EMVs is low, if we directly use $Q_E$ to control traffic signals, we might encounter the issue of congestion among RVs. Consequently, we use $Q_N$ to control the traffic signals and learn $Q_E$ using $\mathcal{L}_{\text{EMV}}$ as presented in Eq. 13.

$$\mathcal{L}_{\text{EMV}} = \text{MSE}\left(Q_E\left(s, \text{EpsRand}\left(\arg\max_{a'}\left(Q_N(s,a')\right), \epsilon\right)\right), \boldsymbol{r}^e + \gamma \max_{a'} Q_N(s',a')\right) \quad (13)$$

In joint training stage, both $Q_N$ and $Q_E$ have acquired a reasonable strategy. However, $Q_E$ is not yet optimized for EMVs. Therefore we perform $\epsilon_t$-decay to select action $a$, which is defined as $a = \text{EpsRand}\left(\arg\max_{a'}\left(Q_N(s,a') + Q_E(s,a') \cdot \mathbf{1}_{\{u < 1-\epsilon_t\}}\right), \epsilon\right)$, here $\epsilon_t$ is a decaying parameter that starts at 1 and decays to 0, $\mathbf{1}$ is an indicator function, and $u \sim \mathcal{U}(0,1)$. This approach enables $Q_E$ to transfer its $Q$-values to those that can influence the traffic signal. Then $\mathcal{L}_{\text{Joint}}$ is defined as in Eq. 14.

$$\mathcal{L}_{\text{Joint}} = \text{MSE}\left(Q_N(s,a) + Q_E(s,a), \boldsymbol{r}^n + \boldsymbol{r}^e + \gamma \max_{a'}\left(Q_E(s',a') + Q_N(s',a')\right)\right) \quad (14)$$

By adopting the aforementioned multi-stage training strategy, we can obtain $Q_N$ that is identical to RV-prioritized methods, and $Q_E$ that is dedicated to optimizing for EMVs. Subsequently, we can utilize ASM described in Sec. 3.2 to derive the final $Q$ function, as depicted on the right of Fig. 2.

## 4 EXPERIMENTS

### 4.1 EXPERIMENTAL SETUP

**Baselines** We compare SplitEMV with a range of representative baselines, which can be categorized into two groups: traditional TSC methods and deep reinforcement learning based TSC methods. The

traditional methods include FixedTime, MaxBandLittle et al. (1981), SOTLCools et al. (2013), Max-PressureVaraiya (2013), and MARLIN-ATSCEl-Tantawy et al. (2013). The DRL-based approaches contain CoLightWei et al. (2019b), PressLightWei et al. (2019a), MPLightChen et al. (2020), and MVNXu et al. (2023). Specifically, EMVLight Su et al. (2022; 2023) prioritizes EMVs through position-aware reward mechanisms. Further details on these methods are provided in Appendix K.

**Datasets and Environments**   We evaluate our approach on two widely used multi-intersection TSC datasets: the Hangzhou (HZ) and Jinan (JN) datasets Wei et al. (2019b), each containing one hour of traffic flow data. To support EMV-aware control, we extend both datasets by labeling certain vehicles as EMVs. During the training, EMVs are randomly selected, and during the testing vehicles whose IDs are divisible by 1000 are designated as EMVs. The driving strategy of these EMVs are identical to RVs but are tracked separately to evaluate EMV passage efficiency.

**Evaluation Metrics**   We adopt the Average Travel Time (ATT) as the primary evaluation metric. This metric is widely employed to assess the performance of TSC. It computes the average time that all vehicles spend from entering to leaving the traffic network during the simulation, which is formulated as $\text{ATT} = \frac{1}{N} \sum_{i=1}^{N} \left( t_i^l - t_i^e \right)$, where $N$ represents the total number of vehicles, and $t_i^e, t_i^l$ are the entering and leaving time of the $i$-th vehicle. A lower ATT indicates superior performance.

**Implementation Details**   All experiments are conducted in SUMO Lopez et al. (2018) simulator. To ensure stability, we repeat each experiment multiple times and report the mean and variance. The implementation details and hyper-parameters can be found in Appendix L. We evaluate the performance of methods with Average Travel Time of RVs and EMVs in seconds.

## 4.2 COMPARISON RESULTS

Table 1: Overall performance of different methods, reported as average travel time in seconds.

| Method Type | Methods | JN | | HZ | |
|---|---|---|---|---|---|
| | | RVs(s) | EMVs(s) | RVs(s) | EMVs(s) |
| Traditional Methods | FixedTime | 442.91 | 222.69 | 541.72 | 1070.9 |
| | MaxBand | 359.38 | 134.95 | 443.86 | 444.03 |
| | SOTL | 386.59 | 131.04 | 420.52 | 243.90 |
| | MaxPressure | 394.58 | 290.51 | 380.19 | 120.13 |
| | MARLIN | 383.60 | 125.23 | 392.14 | 393.49 |
| Deep RL Methods | CoLight | 333.11±2.89 | 148.41±3.23 | 349.01±.172 | 78.038±14.9 |
| | PressLight | 334.01±1.53 | 160.72±21.5 | 357.83±.981 | 112.29±9.12 |
| | MPLight | 348.02±1.78 | 126.35±14.0 | 353.00±2.27 | 123.20±37.6 |
| | MVN | 436.82±34.5 | 491.23±12.9 | 445.74±6.17 | 173.66±104 |
| | EMVLight | 435.22±7.36 | 130.39±.783 | 385.90±2.78 | 186.31±41.8 |
| | SplitEMV | 333.18±1.12 | **38.161±1.28** | 345.77±.468 | **48.960±3.02** |

**Overall Performance Comparison**   We first evaluate the overall performance of all baseline methods on both RVs and EMVs. Tab. 1 reports the average travel time and standard deviation across multiple runs on the HZ and JN datasets. For traditional methods, results are deterministic and thus no standard deviation is reported.

From the results, we observe that Deep RL methods generally outperform traditional methods in terms of RV efficiency. However, when considering EMVs, only EMVLight and our proposed SplitEMV are specifically designed for EMV-aware control. While EMVLight incorporates EMV-specific designs, its lower efficiency for RVs leads to limited overall benefits.

In contrast, SplitEMV achieves the lowest average travel time for EMVs across all settings while maintaining competitive performance for RVs, which is on par with or better than state-of-the-art DRL baselines. Other DRL methods, which are not optimized for EMV scenarios, exhibit high variance in EMVs performance, indicating their instability and limited robustness in this setting.

We also analyze the performance of existing methods when trained with weighted rewards. Due to the space constraints, the detailed results are put in Appendix M. These results highlight a clear trade-off:

small $\beta$ values may offer marginal improvement in the performance of EMVs without severely harming regular vehicle flow, but large $\beta$ values lead to a significant drop in RV performance, which also leads to the increase of average travel time of EMVs. This confirms the difficulty of achieving joint optimization via weighted rewards. The analysis of training and inference time provided in Appendix P. Our method shows no significant difference in runtime compared with other methods.

### 4.3 ABLATIONS

**The effectiveness of Decoupled Learning and ASM**  Tab. 2 presents the experimental results of training with a weighted reward function instead of using Decoupled Learning and ASM. The superscript in the name indicates the weight coefficient $\beta$. We can observe that when $\beta$ is small, the method fails to significantly improve the traffic efficiency for EMVs. Conversely, when $\beta$ is large, although EMV efficiency may improve, the performance for RVs drops significantly, and EMV efficiency is still suboptimal. In contrast, Decoupled Learning and ASM achieves excellent performance for both RVs and EMVs.

Table 2: Comparison of SplitEMV with different weighted reward, loss function and training stages.

| Methods | JN | | HZ | |
|---|---|---|---|---|
| | RVs(s) | EMVs(s) | RVs(s) | EMVs(s) |
| $\text{SplitEMV}^0$ | 332.44±.218 | 60.000±5.77 | 345.13±.532 | 113.33±43.2 |
| $\text{SplitEMV}^{0.1}$ | 332.60±1.22 | 42.022±.711 | 349.15±.219 | 77.234±1.05 |
| $\text{SplitEMV}^{0.3}$ | 332.94±1.69 | 51.099±8.44 | 345.64±.137 | 54.952±1.07 |
| $\text{SplitEMV}^1$ | 340.98±.688 | 43.356±1.33 | 367.68±3.23 | 51.097±1.15 |
| $\text{SplitEMV}^3$ | 410.67±42.0 | 47.976±2.84 | 412.74±9.43 | 57.046±4.76 |
| $\text{SplitEMV}^{10}$ | 505.46±31.0 | 55.947±8.67 | 394.55±13.4 | 251.06±175 |
| $\text{SplitEMV}_{\text{combine}}$ | 339.66±1.51 | 36.805±2.72 | 369.94±5.35 | 425.00±373 |
| $\text{SplitEMV}_{\text{split}}$ | 336.46±.943 | 38.878±.929 | 372.68±12.7 | 285.02±242 |
| $\text{SplitEMV}_{\text{RV}}$ | 332.52±1.36 | 112.92±22.4 | 345.54±.347 | 81.806±3.90 |
| $\text{SplitEMV}_{\text{EMV}}$ | 352.10±10.5 | 40.420±3.95 | 372.83±10.9 | 49.466±4.98 |
| SplitEMV | 333.18±1.12 | **38.161±1.28** | 345.77±.468 | **48.960±3.02** |

**Settings of Different Loss Functions**  We evaluate our method on Hangzhou and Jinan scenarios with multiple loss function selections, analyzing their strengths and limitations. Ultimately, we adopt a three-stage training strategy with distinct loss functions for each stage. Tab. 2 presents the experimental results under different loss function designs, as well as the performance of the model after each individual training stage. Here, $\text{SplitEMV}_{\text{combine}}$ represents the results obtained using the combined loss $\mathcal{L}_{\text{combine}}$ throughout training, while $\text{SplitEMV}_{\text{split}}$ refers to training with the split loss $\mathcal{L}_{\text{split}}$. $\text{SplitEMV}_{\text{RV}}$ and $\text{SplitEMV}_{\text{EMV}}$ correspond to the outcomes after the first and second stages of the multi-stage training, respectively. Since the $\text{SplitEMV}_{\text{RV}}$ focuses solely on optimizing traffic efficiency of RVs, the $Q$-function for EMVs is omitted during evaluation.

From Tab. 2, it is evident that our proposed multi-stage training strategy, along with stage-specific loss functions, significantly improves the ability of model to distinguish and optimize for the two control objectives compared to using a single loss function ($\mathcal{L}_{\text{combine}}$ or $\mathcal{L}_{\text{split}}$). We can also find each stage in has a clearly defined objective: after RV model learning stage, the model performs well for RVs; after EMV model initialization stage, the model adopts a fair strategy to optimize both RVs and EMVs simultaneously; finally after joint training stage, the model further explores optimal control strategies based on the current state, surpassing the performance of the greedy strategy from Stage 2 and achieving further improvements. We further conduct ablation studies on communication method and different EMV appearance rate. The corresponding results are presented in Appendix N and O.

### 4.4 ZERO-SHOT EMV GENERALIZATION ON EXISTING METHODS

As analyzed in Sec. 3.1, Decoupled Learning separates the control strategies for RVs and EMVs. Therefore, the trained EMV-aware TSC policy can be directly integrated with other Q-learning-based regular TSC policies. We integrate the trained RV and EMV prioritized strategy using ASM in a zero-shot fashion with various existing methods to evaluate its generalization capability. The experimental results are presented in Tab. 3. The symbol "+" indicates that EMV model is merged

Table 3: Zero-shot EMV generalization on existing methods in HZ and JN datasets.

| Methods | JN | | HZ | |
|---|---|---|---|---|
| | RVs(s) | EMVs(s) | RVs(s) | EMVs(s) |
| CoLight | 333.11±2.89 | 136.08±14.3 | 349.01±.172 | 94.988±1.15 |
| CoLight+ | 357.72±13.7 | 41.938±5.11 | 375.27±12.7 | 57.360±.592 |
| CoLight* | 334.01±2.74 | **35.374±.142** | 349.19±.698 | **51.652±4.93** |
| PressLight | 334.01±1.53 | 120.77±1.64 | 357.83±.981 | 97.122±8.25 |
| PressLight+ | 335.50±1.96 | 46.356±3.31 | 359.36±1.71 | 55.582±3.12 |
| PressLight* | 336.32±.348 | **41.666±4.40** | 358.39±1.25 | **50.339±3.88** |
| MPLight | 346.46±1.35 | 135.00±27.9 | 353.52±2.79 | 123.06±29.5 |
| MPLight+ | 348.99±.040 | 44.579±5.82 | 356.30±.89 | 81.471±22.6 |
| MPLight* | 347.89±.638 | **43.101±.473** | 353.40±2.11 | **80.602±17.7** |
| MVN | 440.13±31.7 | 312.91±172 | 447.70±4.94 | 144.20±26.7 |
| MVN+ | 537.74±24.0 | 181.28±141 | 483.27±71.0 | 68.993±13.3 |
| MVN* | 433.11±26.6 | **154.90±106** | 428.74±11.2 | **50.873±.935** |

with $\beta = 1$, while "*" indicates the use of the ASM method. Note that EMVLight is excluded from this experiment as it does not independently optimize RVs but instead jointly optimize RVs and EMVs. From the results, we observe that the integration strategy preserves the performance of original methods on RVs while significantly improving the efficiency for EMVs.

## 5 RELATED WORK

**Emergency Vehicle Optimization**    Existing researches on EMV optimization have explored strategies aimed at route optimization Lu & Wang (2019); Kwon et al. (2003); Humagain et al. (2020). Other methods Nelson & Bullock (2000); Qin & Khan (2012); Huang et al. (2015); Bieker-Walz & Behrisch (2019); Wu et al. (2020) have also probed into traffic signal preemption strategies to tackle the issue that RVs may not be able to yield EMVs. Nonetheless, existing studies predominantly center around either route planning for EMVs or pre-set adjustments to traffic signals. EMVLight Su et al. (2022; 2023) integrates real-time route planning with an adaptive TSC algorithm that gives precedence to EMVs. Nevertheless, its real-time planning system exhibits limited effectiveness, and the multi-level signal architecture further exacerbates the training complexity and makes it difficult to transfer to other scenarios. This approach tightly couples the optimization of EMVs and RVs, affecting the strategy for regular TSC problems.

**Deep Reinforcement Learning in TSC**    Deep reinforcement learning have made great progress in Traffic Signal Control in the last ten years. Here, we briefly review the latest advances in traditional TSC methods Buckley & Wheeler (1964); Hunt et al. (1982); Lowrie (1990); Cools et al. (2013); Varaiya (2013) and RL-based TSC methods Mnih et al. (2015); van Hasselt et al. (2016); Wang et al. (2016); Fortunato et al. (2018); Schaul et al. (2016); Mnih et al. (2016); Schulman et al. (2017); Haarnoja et al. (2018); Horgan et al. (2018); Espeholt et al. (2018); Rashid et al. (2018), which are related to our work. Detailed related works are available in the Appendix A.

## 6 CONCLUSION

In this paper, we propose a decoupled policy fusion framework to address the limitations of single-objective reward designs in the presence of EMV-aware TSC problem. By combining independently learned policies for RVs and EMVs with Adaptive Strategy Merging, our method mitigates approximation errors in $Q$-function learning and enhances policy performance. In addition, a new agent architecture **SplitEMV** is introduced to improve inter-agent communication. Experiments on real-world data show that our method and model significantly improves efficiency of EMVs without compromising RVs. Moreover, the EMV policy can be combined with existing control methods in a zero-shot manner, offering strong generalization and practical applicability.

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

APPENDIX

## A MORE RELATED WORKS

**Traffic Signal Control** TSC methods have been extensively studied and widely deployed to mitigate congestion and improve traffic flow. Classic approaches include Fixed-time Buckley & Wheeler (1964), SCOOT Hunt et al. (1982), and SCATS Lowrie (1990), which rely on pre-defined timing plans or centralized rule-based adaptations to traffic conditions. These systems are stable but struggle to adapt to complex urban traffic. Later efforts such as SOTL Cools et al. (2013) and Max-Pressure Varaiya (2013) attempt to improve responsiveness with local traffic states. Nevertheless, they lack the flexibility to generalize across diverse intersection topologies. Recently, RL-based TSC methods are widely researched. Initial works on single-intersection TSC problem such as Intellilight Wei et al. (2018), FRAP Zheng et al. (2019), and AttendLight Oroojlooy et al. (2020), are struggle to apply on multiple intersection due to the partially observed states. To improve the performance on multi-intersection TSC problem, methods containing centralized agent A. & Bhatnagar (2011), communication Wei et al. (2019b), advanced reward Wei et al. (2019a); Chen et al. (2020), GNN Nishi et al. (2018), partitioning Ma & Wu (2024) and hierarchical learning Zhu et al. (2022). However, their dependence on manually tuned parameters often limits their effectiveness, and they are unable to improve the efficiency of EMVs.

**Deep Reinforcement Learning** To address the limitations of traditional RL in large state spaces, deep reinforcement learning (DRL) leverages neural networks to approximate value and policy functions. A representative method is DQN Mnih et al. (2015), which introduces target networks and experience replay. Subsequent works improve stability and exploration through techniques van Hasselt et al. (2016); Wang et al. (2016); Fortunato et al. (2018); Schaul et al. (2016) such as double $Q$-learning, dueling architecture, noisy networks, and prioritized replay. In parallel, actor-critic frameworks Mnih et al. (2016); Schulman et al. (2017); Haarnoja et al. (2018) are widely adopted for continuous control and stable policy optimization. Large-scale extensions including Ape-X, IMPALA, and multi-agent methods Horgan et al. (2018); Espeholt et al. (2018); Rashid et al. (2018) further broaden DRL's applicability in complex and cooperative environments such as traffic systems.

## B DEC-POMDP MODELING FOR EMV-AWARE TSC PROBLEM

We model the TSC task as a Dec-POMDP, defined as

$$G = \langle \mathcal{S}, \mathcal{A}, P, r, \mathcal{Z}, O, N, \gamma \rangle$$

where $\boldsymbol{s} \in \mathcal{S}$ is the true global state of the environment. $N$ denotes the number of agents, each controlling one intersection. Each agent $i \in \{1, \ldots, N\}$ selects an action $a_i \in \mathcal{A}$. The joint action is denoted as $\boldsymbol{a} = [a_i]_{i=1}^N \in \mathcal{A}^N$. The environment transitions according to $P(\boldsymbol{s}'|\boldsymbol{s}, \boldsymbol{a}) : \mathcal{S} \times \mathcal{A}^N \times \mathcal{S} \to [0, 1]$. The environment returns a joint reward $r(\boldsymbol{s}', \boldsymbol{s}, \boldsymbol{a}) : \mathcal{S} \times \mathcal{A}^N \times \mathcal{S} \to \mathbb{R}$. Each agent receives a local observation $\boldsymbol{z}_i = O(\boldsymbol{s}, i) \in \mathcal{Z}$, where $\boldsymbol{z}_i \subseteq \boldsymbol{s}$. The goal of each agent is to learn a policy $\pi_i^*$ that maximizes the global discounted cumulative reward

$$\sum_{t=0}^{\infty} \gamma^t r(\boldsymbol{s}_t, \boldsymbol{a}_t)$$

with the discount factor $\gamma \in [0, 1]$.

To solve the EMV-Aware TSC problem, it is modeled as a Dec-POMDP with the following components:

- **State space $s$.** At time $t \in \mathbb{N}$, the true state is $\boldsymbol{s}^t = \langle \mathcal{I}, \mathcal{R}, \boldsymbol{P}^t, \boldsymbol{V}^t, \boldsymbol{E}^t \rangle$, where $\boldsymbol{P}^t = [P_i^t]_{i=1}^N$ denotes the current phases of all intersections, $\boldsymbol{V}^t = [V_i^t]_{i=1}^M$ the states of all vehicles, and $\boldsymbol{E}^t = [E_i^t]_{i=1}^P$ the states of emergency vehicles. We assume that each intersection has at most one active emergency vehicle at a time: $|E_i^t| \leq 1$.

- **Observation $z$.** Each agent $A_i$ observes a partial view $z_i^t = O(\boldsymbol{s}^t, i) = \langle \mathcal{I}, \mathcal{R}, P_i^t, \boldsymbol{v}_i^t, \boldsymbol{E}^t \rangle$. Due to sensor limitations, the agent cannot access detailed vehicle-level information. Instead, it receives aggregated statistics (e.g., vehicle counts per lane) via sensors like magnetic loops or cameras. In

contrast, emergency vehicles are assumed to be network-connected, allowing full access to their state (location, velocity, route).

• **Action space $a$.** After receiving observation $z_i^t$, agent $A_i$ selects an action $a_i^t \in \mathcal{A}_i$, representing the next signal phase for duration $\Delta t$. If the selected phase $P_i^{t+1}$ differs from the current phase $P_i^t$, a transition period of 5 seconds is introduced to ensure the safe passage of vehicles through the intersection. During this transition period, vehicles from all directions are prohibited from entering the intersection.

• **Joint reward function $r$.** To balance traffic efficiency and emergency response, we design a joint reward that decomposes into local rewards at each intersection. Each local reward includes two components: one for regular vehicles and one for emergency vehicles. Following the unbiased estimation method proposed by Hua et al. Wei et al. (2019b), the average number of vehicles on incoming lanes is used as an unbiased estimator of average travel time. For emergency vehicles, we directly count their presence on incoming lanes. The reward function at intersection $i$ is:

$$r_i^t = r_i^{n,t} + \beta r_i^{e,t} \tag{15}$$

where $\beta$ controls the weight of emergency vehicle priority. Finally, the global joint reward is the sum of all local rewards in intersection $i$.

## C  BASIC DEFINITION OF TSC PROBLEM

**Definition C.1** (Intersection). *An **intersection** $I_i \in \mathcal{I}$ denotes the endpoint or starting point of one or more roads. Each intersection is controlled by a traffic signal. We assume each intersection has exactly four incoming and four outgoing roads.*

**Definition C.2** (Road). *A **road** $R_{i,j} \in \mathcal{R}$ is a directed edge from intersection $I_i$ to intersection $I_j$. Let $\mathcal{R}$ be the set of all roads. Each road consists of three lanes: left-turn ($L_{i,j}^l$), straight ($L_{i,j}^s$), and right-turn ($L_{i,j}^r$).*

**Definition C.3** (Traffic Signal Phase). *A **phase** $P_i$ at intersection $I_i$ defines a subset of allowed traffic movements from incoming roads to outgoing roads. If a movement is allowed in a phase, vehicles planning to take that movement can proceed; otherwise, they must stop and wait. Let $\mathcal{A}_i$ denote the set of all legal signal phases (i.e., the action space for agent $A_i$).*

**Definition C.4** (Vehicle). *A **vehicle** $V_i \in \mathcal{V}$ enters the road network from its boundary and travels along a predefined route until it exits. Let $\mathcal{V}$ denote the set of all vehicles. A vehicle's state is represented as:*

$$V_i = \langle L_{x,y}^z, s, v, T \rangle$$

*where $L_{x,y}^z$ is the current lane, $s$ is the position on the lane, $v$ is the velocity, and $T$ is the planned route.*

## D  $\beta$ SENSITIVITY IN EMV-AWARE TSC PROBLEM

We make a detailed discussion about what will affect for different $\beta$. We firstly give the detailed definition of the simplified single intersection traffic signal control problem. Then we consider the error distribution when we use neural networks to approximate $Q$ function. Finally we analyze the effect of the number of vehicles on the $Q$ function.

### D.1  SIMPLIFIED SINGLE INTERSECTION EMV-AWARE TSC PROBLEM

The simplified single-intersection EMV-aware TSC problem is defined as follows: Assume that there are four incoming roads at the intersection, and each road has only one incoming lane. All vehicles will leave directly from the opposite lane after passing through the intersection. There are four phases at the intersection, each phase corresponds to one of the four incoming lanes. At the initial state, there are $k$ vehicles on each lane, when the time passes $\Delta t$, a random lane will have $p$ vehicles entering, and the signal light needs to select a phase, which will allow at most $p$ vehicles to leave the intersection on that corresponding lane, and we assume $k \geq p$. At a random time $t$, one of the incoming vehicles will be an emergency vehicle.

### D.2 VARIANCE INTRODUCED BY FUNCTION APPROXIMATION

Consider approximating $Q_N$, when using a neural network for approximation, as the gradient descent method is used to fit the function, the trained $Q_N$ can be approximately expressed as:

$$Q_N(s,a) \sim \mathcal{N}(Q_N^*(s,a), \sigma^2) \tag{16}$$

where $Q_N^*(s,a)$ is the true value of $Q_N(s,a)$, $\sigma$ is the standard deviation, which indicates that $Q_N(s,a)$ is unbiased with respect to the true value $Q_N^*(s,a)$, but the final learned result has a standard deviation due to the error of gradient descent. When considering the traffic signal control problem, the reward is positively correlated with the number of vehicles, as a result, if the number of vehicles in state $s'$ is enlarged by $k$ times, the corresponding $Q_N^*(s',a)$ will also be enlarged by $k$ times. This means that we can treat $Q_N^*$ satisfies homogeneity, i.e., $\lambda f(x) = f(\lambda x)$. For neural networks, when trained with data that satisfies homogeneity, it can learn the homogeneity feature approximately. Then the standard deviation $\sigma$ will also be enlarged by $k$ times. This standard deviation will cause $Q_N$ and $Q_E$ to interfere with each other, make the inequality Eq.7 not hold, and ultimately affect the selection of the optimal action.

### D.3 EFFECT OF AVERAGE VEHICLE NUMBER

For the proposed environment, We can find an optimal strategy that minimize the average driving time for regular vehicles, i.e., select the phase corresponding to the lane that has the most vehicles waiting. With this strategy, the average number of vehicles on the lane is always $k$, therefore the reward at each time is $-k$. Considering the discount factor $\gamma \in (0,1)$, the $Q_N^*$ function of this strategy is:

$$Q_N^*(s,a) = \sum_{t=0}^{\infty} \gamma^t(-k) = -k\sum_{t=0}^{\infty} \gamma^t = -k\frac{1-\gamma^\infty}{1-\gamma} = \frac{k}{\gamma-1} \tag{17}$$

Based on the above analysis, we can find that when using neural network to approximate $Q_N^*(s,a)$, we have:

$$Q_N(s,a) \sim \mathcal{N}(Q_N^*(s,a), \sigma_N^2) = \mathcal{N}\left(\frac{k}{\gamma-1}, (k\sigma_N)^2\right) \tag{18}$$

where $\sigma_N$ is the standard deviation of $Q_N$, which is positively correlated with $Q_N^*(s,a)$, i.e., when $k$ is enlarged, average and standard deviation of $Q_N(s,a)$ will also be enlarged.

Then we consider $Q_E$. Without loss of generality, we assume that the emergency vehicle appears on the lane in the east direction at time $t=0$. The optimal strategy to minimize the average driving time of the emergency vehicle is to keep the east lane open until the emergency vehicle leaves the intersection at time $t' = \left\lceil \frac{k}{p} \right\rceil$. Then we have:

$$Q_E^*(s,a) = \sum_{t=0}^{t'-1} \gamma^t(-1) = -\sum_{t=0}^{t'-1} \gamma^t = -\frac{1-\gamma^{t'}}{1-\gamma} = \frac{1-\gamma^{t'}}{\gamma-1} \tag{19}$$

$$Q_E(s,a) \sim \mathcal{N}(Q_E^*(s,a), \sigma_E^2) = \mathcal{N}\left(\frac{1-\gamma^{t'}}{\gamma-1}, \sigma_E^2\right) \tag{20}$$

As $k$ has negligible effect on $Q_E^*$, we can assume that $\sigma_E$ is a constant.

Consider Proposition 3.1, we define:

$$Q_N^\Delta = \max(Q_N) - \min(Q_N) \tag{21}$$

$$Q_N^\delta = \max(Q_N) - \text{second\_max}(Q_N) \sim \mathcal{N}(kC_N, (k\sigma_N)^2) \tag{22}$$

$$Q_E^\Delta = \max(Q_E) - \min(Q_E) \tag{23}$$

$$Q_E^\delta = \max(Q_E) - \text{second\_max}(Q_E) \sim \mathcal{N}(C_E, \sigma_E) \tag{24}$$

where $C_N$ and $C_E$ are constants, and $Q_N$ will increase with the increase of $k$, but $Q_E$ is independent of $k$. Therefore, we have:

- When we treat $Q_E$ as $Q_1$ in Eq. 7, we have:

$$Q_E^\delta > Q_N^\Delta$$
$$C_E + \sigma'_E > kC_N + k\sigma'_N \geq k\sigma'_N$$
$$C_E > \sigma'_E + k\sigma'_N \qquad (25)$$

where $\sigma'_N$ and $\sigma'_E$ are the standard deviations of $Q_N$ and $Q_E$, respectively, caused by model training. As the standard deviation is equally likely to be positive or negative when sampling from a normal distribution, the negative sign can be reverted when we move $\sigma'_E$ to the right side of the inequality. Therefore, when $k$ is too large, the interference caused by the training standard deviation on the right side of Eq. 25 will increase, making Eq. 7 not hold, and the maximum value of $Q$ may be different from that of $Q_E$.

- When we treat $Q_N$ as $Q_1$ in Eq. 7, we have:

$$Q_N^\delta > Q_E^\Delta$$
$$kC_N + k\sigma'_N > C_E + \sigma'_E \geq \sigma'_E$$
$$C_N > \frac{\sigma'_E}{k} + \sigma'_N \qquad (26)$$

Therefore, when $k$ is too small, the interference caused by the training standard deviation on the right side of Eq. 26 will increase, making Eq. 7 not hold, and the maximum value of $Q$ may be different from that of $Q_N$.

To summarize, when $Q_N$ and $Q_E$ are fixed weighted, $k$ can only be guaranteed to be in a certain range to ensure that Eq. 7 holds, and it cannot handle the full range interval.

## E  NUMERICAL DISTRIBUTION OF $Q_N$ AND $Q_E$

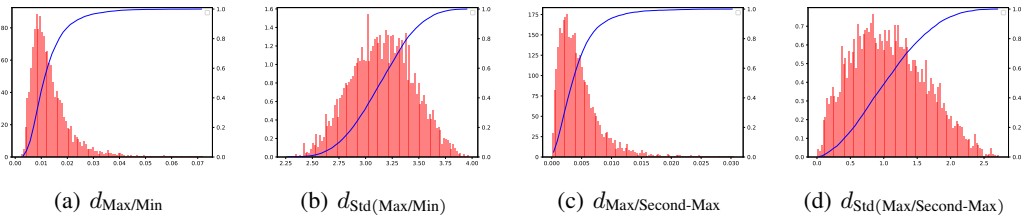

| (a) $d_{\text{Max/Min}}$ | (b) $d_{\text{Std(Max/Min)}}$ | (c) $d_{\text{Max/Second-Max}}$ | (d) $d_{\text{Std(Max/Second-Max)}}$ |

Figure 3: Numerical Distribution of all $Q_N(s,a)$ during one epoch testing process. $d_{A/B}$ means the difference between value A and value B, where A and B can be maximum value (Max), second-maximum value (Second-Max), or minimum value (Min). $d_{\text{Std}(A/B)}$ is the difference between value A and value B, but firstly applying normalization on $Q_N(s,a)$ across all actions.

Fig. 3 and Fig. 4 illustrate the numerical distributions of $Q_N$ and $Q_E$ trained in a real-world scenario without and with normalization. As shown in Fig. 3(b) and Fig. 4(b), the normalized distribution of $Q_N$ approximates a Gaussian distribution, while the distribution of $Q_E$ is concentrated near zero, with only a few larger values. This observation aligns with our earlier assumption: $Q_N$ follows a near-normal distribution, whereas $Q_E$ remains tightly clustered around zero.

Further analysis in Fig. 3(b)(d) and Fig. 4(b)(d) reveals that, after normalization using Eq. 8 and Eq. 9, the $d_{\text{Std(Max/Min)}}$ for $Q_E$ values close to zero is generally less than 0.3. Additionally, for $Q_N$, fewer than 5% of the samples violate Eq. 7 when considering $d_{\text{Std(Max/Second-Max)}}$. In contrast, without normalization as shown in Fig. 3(a)(c) and Fig. 4(a)(c), the difference between the maximum and minimum values of $Q_N$ is less than 0.1, making the maximum of $Q_N$ highly susceptible to being influenced by values from $Q_E$.

It is also worth noting that Eq. 7 captures the worst-case scenario: when the best action according to $Q_E$ corresponds to the second-best action under $Q_N$, and the worst action under $Q_E$ coincides with the best action under $Q_N$. In practice, when $d_{\text{Std(Max/Second-Max)}}$ of $Q_N$ is small, the concentrated distribution of $Q_E$ is unlikely to alter the optimal action selection based on $Q$.

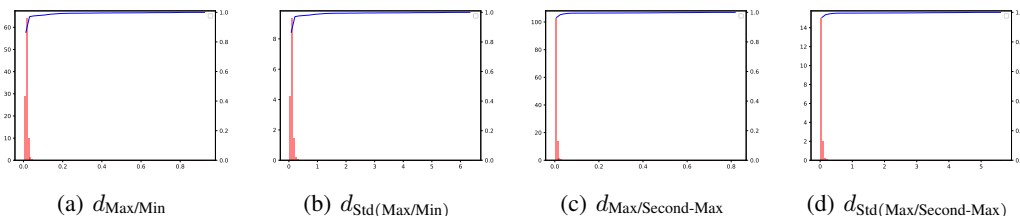

(a) $d_{\text{Max/Min}}$      (b) $d_{\text{Std(Max/Min)}}$      (c) $d_{\text{Max/Second-Max}}$      (d) $d_{\text{Std(Max/Second-Max)}}$

Figure 4: Numerical Distribution of all $Q_E(s,a)$ during one epoch testing process. $d_{\text{A/B}}$ means the difference between value A and value B, where A and B can be maximum value (Max), second-maximum value (Second-Max), or minimum value (Min). $d_{\text{Std}(A/B)}$ is the difference between value A and value B, but firstly applying normalization on $Q_E(s,a)$ across all actions.

Table 4: Performance under different values of $\beta$ during the one-epoch simulation.

| $\beta$ | 0.1 | 0.5 | 1 | 5 | 10 |
|---|---|---|---|---|---|
| **JN-RV** | 332.8±1.0 | 334.0±0.3 | 333.2±1.1 | 333.0±0.7 | 331.8±0.7 |
| **JN-EMV** | 38.3±0.8 | 37.1±2.3 | 38.2±1.3 | 37.9±0.6 | 37.2±0.9 |
| **HZ-RV** | 346.0±0.7 | 345.9±0.8 | 345.8±0.5 | 346.0±0.5 | 346.5±0.9 |
| **HZ-EMV** | 49.0±0.2 | 49.4±1.2 | 49.0±3.0 | 50.0±3.0 | 50.3±0.6 |

## F    STABILITY OF ONE-EPOCH SIMULATION

We performed one-epoch simulation to estimate the statistics $\sigma_E$ on the Q-function of the learned EMV policy, rather than to artificially interfere with the characteristics of the test scenarios.

We can obtain $\sigma_E$ with any scenario, because it is closely related to the policy itself and has very minor relation to the test scenarios. For example, for a same model, the value of $\sigma_E$ obtained through replay in JN is $0.154$, and in HZ it is $0.145$. When the frequency of EMV occurrence changes from $\frac{1}{100}$ to $\frac{1}{5000}$, the value range of $\sigma_E$ is between $0.137$ and $0.159$. The results highly support that no matter how the test scenarios change, the obtained $\sigma_E$ of a same model is very stable.

The purpose of one-epoch simulation is to avoid manual parameter tuning of $\sigma_E$ and enable the model to get this parameter adaptively. In practical applications, we can estimate $\delta_E$ in the simulation system based on real-vehicle data and then directly apply it to real-world scenarios.

During one-epoch simulation, we set the default $\beta$ as 1, because in Joint Training Stage, we use $\beta = 1$ to sum up Q-values of two strategies. However, Adaptive Strategy Merging is insensitive to the value of $\beta$ selected in the simulation. In Table 4, we show the performance when we use different $\beta$ in the one-epoch simulation to calculate $\sigma_E$. We can find the performance is very stable on both datasets even $\beta$ increases or decreases 10 times.

## G    PROOFS FOR PROPOSITIONS

### G.1    PROOF OF PROPOSITION 3.1

*Proof.* As $Q = Q_1 + Q_2$, we denote the maximum element position of $Q_1$ as $i$ and the second largest element position as $j$. Then, we can transform the inequality 7 into:

$$\max(Q_1) - \text{second\_max}(Q_1) > \max(Q_2) - \min(Q_2) \tag{27}$$

$$\max(Q_1) + \min(Q_2) > \text{second\_max}(Q_1) + \max(Q_2) \tag{28}$$

For $Q_1$, we have:

$$
\begin{aligned}
Q(s, i) &= Q_1(s, i) + Q_2(s, i) \\
&\geq Q_1(s, i) + \min(Q_2) \\
&= \max(Q_1) + \min(Q_2) \\
&> \text{second\_max}(Q_1) + \max(Q_2) \\
&\geq Q_1(s, j) + \max(Q_2) \\
&\geq Q(s, j), \quad \text{for } j \neq i
\end{aligned}
\tag{29}
$$

Therefore, $Q(s, i)$ is the maximum element of $Q$, which means that the position of the maximum element of $Q$ is the same as that of the maximum element of $Q_1$. $\qquad\square$

### G.2 PROOF OF PROPOSITION 3.2

*Proof.* Assume that the optimal action of $Q$ is $a$, then for $Q'$ we have:

$$
\begin{aligned}
Q'(s, a) &= \frac{Q(s, a) - \mu}{\sigma^2} \\
&> \frac{Q(s, j) - \mu'}{\sigma^2} \\
&= Q'(s, j), \quad \text{for } j \neq a
\end{aligned}
\tag{30}
$$

Therefore $Q'(s, a)$ is the maximum element of $Q'$, which means that $Q'$ and $Q$ have the same optimal action. $\qquad\square$

## H OPTIMALITY OF DECOUPLING CONTROL STRATEGY

When we define the cumulative reward as:

$$
Q(s, a) = Q_N(s, a) + \beta Q_E(s, a)
\tag{31}
$$

If $Q(s, a)$ is the optimal strategy, we have the following equation:

$$
Q^*(s, a) = Q_N^*(s, a) + \beta Q_E^*(s, a)
\tag{32}
$$

$$
\mathbb{E}\left[\boldsymbol{R}_N^Q + \beta \boldsymbol{R}_E^Q\right] = \mathbb{E}\left[\boldsymbol{R}_N^N\right] + \beta \mathbb{E}\left[\boldsymbol{R}_E^E\right]
\tag{33}
$$

$$
\tag{34}
$$

To make the above equation hold, a solution should be:

$$
\boldsymbol{R}_N^Q + \beta \boldsymbol{R}_E^Q = \boldsymbol{R}_N^N + \beta \boldsymbol{R}_E^E
\tag{35}
$$

$$
\boldsymbol{R}_N^Q = \boldsymbol{R}_N^N
\tag{36}
$$

$$
\boldsymbol{R}_E^Q = \boldsymbol{R}_E^E
\tag{37}
$$

As $\boldsymbol{R}_N^Q$, $\boldsymbol{R}_E^Q$, $\boldsymbol{R}_N^N$ and $\boldsymbol{R}_E^E$ are the cumulative rewards based on three different strategies, if $Q^*(s, a)$, $Q_N^*(s, a)$ and $Q_E^*(s, a)$ are the same strategy, which has the same maximum action for all states, then Eq. 36 and Eq. 37 hold, and $Q(s, a)$ is identical to $Q^*(s, a)$.

## I FAILURE CASES OF ADAPTIVE STRATEGY MERGING

We show a detailed failure case of Adaptive Strategy Merging. Taking the situation in HZ dataset as an example, a wrong case without ASM is given here: The normalized $Q'$ values are as follows:

$$
Q'_N = [0.5352, -0.2903, 1.1374, -0.7531, 1.4310, 0.2416, -1.2328, -1.0689],
$$

$$
Q'_E = [-0.7731, 0.9359, 0.0927, 0.2415, -0.6540, -0.5901, -0.5687, 1.3158].
$$

It can be seen that $Q'_N$ tends to choose action 4, because the lane corresponding to this action has the longest queue. $Q'_E$ tends to choose action 7, and the lane corresponding to this action contains the EMV. However, the finally obtained $Q$ is:

$$Q = [-0.2382, 0.6457, 1.2298, -0.5114, 0.7766, -0.3485, -1.8011, 0.2472]$$

and the final action is 2. This situation is as analyzed in Section 3.2, after adding the two $Q'$ values, the finally selected action is different from the actions preferred by the two policies, thus we consider it as a failure case.

It is difficult to count the occurrence frequency of similar failure cases, as it is hard to determine whether the selected phase is wrong without manually checking. We assume the phases that are different from those of $Q'_N$ and $Q'_E$ as wrong phases, the occurrence frequency of ASM for wrong phases is 2.2%. While when directly using $\beta = 1$, this frequency is 8.4%. This shows that ASM significantly reduces the probability of choosing the wrong phase.

## J    DETAILED DESCRIPTION OF SPLITEMV MODEL

We propose a detailed network structure that is applicable to both $Q_N$ and $Q_E$. The network consists of two main parts: generating the communication information, and estimating the $Q$ function.

The first part of the model is used to generate the communication information. It firstly split the input state into each incoming lane, which contains the number of vehicles on the lane, lane direction, signal phase of the lane, and the speed and position of the emergency vehicle if there is one. Then it use a multi-layer perceptron to generate the lane vector representation $h_i^L$. We then use a multi-head attention module to process the lane vector representation to catch information between lanes.

To make communication information more effective, we consider predicting the contribution of each lane to the corresponding adjacent intersection. It is achieved by predicting the contribution of each lane and group them based on their outgoing directions.

$$h_i^L = \text{MLP}(v_i \oplus d_i \oplus g_i \oplus e_i \oplus s_i) \tag{38}$$

where $v_i$ is the number of vehicles, $d_i$ is the approaching direction of lanes (i.e. go straight, turn left and turn right) with one-hot encoding. $g_i$ indicates whether the current lane is allowed to pass under the overall traffic signal phase $P_i$. $e_i$ indicates whether there is an emergency vehicle, and $s_i$ is the speed of the emergency vehicle, when there is no emergency vehicle, $s_i$ is set to 0. $\oplus$ indicates the vector concatenation operation, and MLP is the multi-layer perceptron, which consists of an input layer, two hidden layers with 32 neurons, and an output layer with ReLU activation function.

Then, the model applies a multi-head attention module to interact and integrate the lane information:

$$\boldsymbol{h}'^L = \text{MultiHeadAttention}(\boldsymbol{h}^L) \tag{39}$$

where $\boldsymbol{h}^L$ is the concatenation of all lane vector representations, and $\boldsymbol{h}'^L$ is the lane information after processing by the multi-head attention module. Then we predict the contribution $g_i$ of each lane with a multi-layer perceptron followed by a Sigmoid function:

$$g_i' = \text{Sigmoid}(\text{MLP}(h_i'^L)) \tag{40}$$

The higher the value of $g_i'$, the higher the contribution that this lane contributes to the adjacent intersection. Finally, we sum up the lane vector representations which are connected to the same intersection, followed by a multi-layer perceptron:

$$v_i^o = \text{MLP}(\sum_{j \in \text{From}(i)} h_j'^L \cdot g_j') \tag{41}$$

where $\text{From}(i)$ is the set of lanes connected to the outgoing lane $i$. $v_o^i$ will be used as the input of the $Q$ function for adjacent intersections.

After receiving the communication information from adjacent agents, the model will combine the communication information with current state as the following, and predict the $Q$ value.

$$h_i = \text{MLP}(v_i \oplus d_i \oplus g_i \oplus e_i \oplus s_i \oplus v_i^o) \tag{42}$$

Compared with Eq. 38, the difference is that we add the predicted traffic flow information $v_i^o$ as the input. Then, for action $a_i$, the corresponding $Q$ function value is:

$$g_i = \text{Permit}(a_i) \tag{43}$$

$$r_i = \text{Reject}(a_i) \tag{44}$$

$$\boldsymbol{h}_i^g = \{h_j | j \in g_i\} \tag{45}$$

$$\boldsymbol{h}_i^r = \{h_j | j \in r_i\} \tag{46}$$

$$p_i = \text{Embedding}(P_i) \tag{47}$$

$$Q_i = \text{MLP}(\overline{\boldsymbol{h}_i^g} \oplus \overline{\boldsymbol{h}_i^r} \oplus p_i) \tag{48}$$

where $\text{Permit}(a_i)$ and $\text{Reject}(a_i)$ are the set of lane numbers that are allowed and rejected to pass the intersection under action $a_i$. $\boldsymbol{h}_i^g$ and $\boldsymbol{h}_i^r$ are the set of lane vector representations that are allowed and rejected to pass the intersection. Embedding is an embedding layer that maps the traffic signal phase $P_i$ into a vector representation. $\overline{\boldsymbol{h}_i^g}$ and $\overline{\boldsymbol{h}_i^r}$ are the average values of $\boldsymbol{h}_i^g$ and $\boldsymbol{h}_i^r$. $Q_i$ is the $Q$ function value corresponding to action $a_i$, i.e. $Q(s, a_i)$.

There are differences when applying the model to different scenarios. When applying the model to $Q_N$, we set $e_i$ and $s_i$ to 0 in Eq. 38 and Eq. 42, to ensure that the model can focus on learning the control logic of regular vehicles. When applying the model to $Q_E$, $e_i$ and $s_i$ will be set according to the actual value.

## K    DETAILS OF COMPARED BASELINES

We make a brief introduction to baseline approaches. **Traditional methods** include:

- **FixedTime**, which based on pre-defined signal phase and time interval to perform cyclic control on traffic signals.
- **MaxBand** Little et al. (1981), which is similar to GreenWave , to control multiple intersections. Each intersection is controlled by FixedTime, and it will change the offset between adjacent intersections to optimize the time on both side to pass the arterial without stopping.
- **SOTL** Cools et al. (2013), which is an adaptive traffic signal control method, it decides whether to change the traffic signal phase base on the vehicle number on lanes that is allowed by current traffic signal phase and next traffic signal phase. It contains two thresholds $g$ and $r$, when vehicle number of current phase is smaller than $g$ and vehicle number of next phase is larger than $r$, the traffic phase is changed.
- **MaxPressure** Varaiya (2013), which proposes the Pressure metric, to indicate whether an intersection has the balanced incoming and outgoing traffic flows, and proposes a greedy method to minimize the Pressure of all intersections.
- **MARLIN** El-Tantawy et al. (2013), which is a tabular Q-learning method to perform multi-intersection traffic signal control, and proposed independent and integrated mode to control agents individually or cooperate with adjacent agents.

**DRL-based methods** include:

- **CoLight** Wei et al. (2019b), which is a multi-agent reinforcement learning method that integrates graph attention for agent communication, and increases the performance compared with traffic signal control methods with individual agents.
- **PressLight** Wei et al. (2019a), which extends from MaxPressure, uses Pressure as the reward function and perform deep reinforcement learning to learn better control strategies that minimizes the Pressure of intersections.
- **MPLight** Chen et al. (2020), which shares the parameters of $Q$ function prediction model between all intersections, integrates the symmetry structure of FRAP Zheng et al. (2019) for better learning speed, and use Pressure as reward function to stablize the training process, which makes it able to control thousands of intersections.
- **MVN** Xu et al. (2023), which is designed to have better robustness under Carlin & Wagner attacks, and performs reward detection to find abnormal rewards from regular rewards, to avoid performing a wrong action when input state is not reliable.

- **EMVLight** Su et al. (2022; 2023), which prioritizes emergency vehicles by designing the path for emergency vehicles to their destination, and can dynamically change in real time based on current road conditions. It also proposes a reinforcement learning based method to control the traffic signals, and designs multi-class reinforcement learning agents to have different reward function based on the position relative to the emergency vehicle.

These baselines provide a comprehensive comparison across classical heuristics, traditional RL, and modern DRL methods with and without emergency vehicle considerations. For algorithms that are not open-sourced, we implemented them ourselves based on the descriptions provided in the corresponding papers. For open-source algorithms that were directly compatible with our research, we used their official implementations. In cases where open-source implementations presented compatibility issues—such as mismatches with the simulator interface or outdated deep neural network frameworks—we re-implemented them in our environment using their released code as reference. Throughout this process, we strictly adhered to the descriptions in the original papers. For parameters not specified in detail, we performed a simple parameter search to determine optimal values.

## L IMPLEMENTATION DETAILS

All experiments are conducted using the SUMO simulator Lopez et al. (2018). Each experiment is initialized with a random seed and trained on a fixed dataset, with periodic validation to identify and retain the best-performing policy. To ensure robustness, each experiment is repeated multiple times, and we report the mean and variance of the results. For open-source methods, we use the official implementations or adopt the hyper-parameters provided in their released code. For non-open-source methods, we follow the hyper-parameters specified in their original papers; for any unspecified parameters, we apply grid search to determine the optimal configuration. To ensure fair comparison, all general hyper-parameters are standardized across methods, as listed in Tab. 5. Hyper-parameters specific to SplitEMV are provided separately in Tab. 6. We conduct all experiments on a system equipped with an AMD Ryzen 7950X CPU, 128 GB DDR5 RAM, and an NVIDIA RTX 4090 GPU, running Ubuntu 20.04, Python 3.10.6, and PyTorch 1.12.0.

Table 5: General hyper-parameters for all methods.

| Parameter | Value |
|---|---|
| Learning Rate | $5 \times 10^{-5}$ |
| Batch Size | 256 |
| Discount Factor $\gamma$ | 0.8 |
| Target Network Update Frequency | 5 |
| Replay Buffer Size | 16000 |
| Initial $\epsilon$-greedy Rate | 0.9 |
| Final $\epsilon$-greedy Rate | 0.02 |
| $\epsilon$-greedy Decay Percentage | 30% |
| Training Steps | 120000 |
| EMV Reward Coefficient $\beta$ | 0.1, 0.3, 1, 3, 10 |
| Training Runs | 4 |
| Validation Runs | 10 |

## M WEIGHTED REWARD ANALYSIS

To further investigate whether existing DRL methods can be adapted for emergency-aware control, we introduce a weighted reward strategy that incorporates a coefficient $\beta$ to balance the priorities between regular and emergency vehicles. For each method, we augment its reward function by assigning additional weight to emergency vehicle rewards, with the corresponding $\beta$ value indicated as a superscript.

Tab. 7 summarizes the performance of each DRL method under varying $\beta$ values, alongside results from the original EMVLight and SplitEMV for comparison. As $\beta$ increases, the efficiency of regular

Table 6: Hyper-parameters specific for SplitEMV.

| Parameter | Value |
|---|---|
| MLP Hidden Layer Number | 2 |
| MLP Hidden Dimension | 32 |
| Communication Hidden Dimension | 32 |
| Stage 1 Training Steps | 60000 |
| Stage 2 Training Steps | 30000 |
| Stage 3 Training Steps | 30000 |

Table 7: Performance with weighted rewards of existing methods (average travel time in seconds, lower is better).

| Reward Function | Methods | JN | | HZ | |
|---|---|---|---|---|---|
| | | RVs | EMVs | RVs | EMVs |
| Weighted Reward | $CoLight^{0.1}$ | 342.53±1.09 | 129.08±7.36 | 359.36±2.46 | 130.31±43.0 |
| | $CoLight^{0.3}$ | 330.85±.417 | 108.61±6.50 | 349.87±2.04 | 77.502±6.91 |
| | $CoLight^1$ | 487.71±48.7 | 286.72±89.9 | 539.87±41.1 | 398.26±218 |
| | $CoLight^3$ | 699.33±66.0 | 342.16±18.6 | 695.77±34.4 | 102.81±45.5 |
| | $CoLight^{10}$ | 767.43±52.2 | 286.20±87.8 | 741.99±89.6 | 326.54±32.6 |
| | $PressLight^{0.1}$ | 336.01±4.48 | 115.08±3.13 | 358.17±.425 | 97.730±4.87 |
| | $PressLight^{0.3}$ | 334.59±.791 | 108.82±1.06 | 359.38±.38 | 132.81±5.39 |
| | $PressLight^1$ | 338.77±1.69 | 110.58±9.99 | 362.23±1.08 | 113.69±17.1 |
| | $PressLight^3$ | 359.81±2.30 | 141.58±28.4 | 372.37±1.29 | 93.141±20.5 |
| | $PressLight^{10}$ | 433.77±10.3 | 150.67±43.0 | 404.11±1.15 | 381.64±221 |
| | $MPLight^{0.1}$ | 354.65±1.12 | 158.25±55.9 | 353.05±4.03 | 151.45±73.8 |
| | $MPLight^{0.3}$ | 350.50±1.72 | 156.80±50.9 | 348.13±1.02 | 92.249±4.21 |
| | $MPLight^1$ | 360.65±12.6 | 194.02±77.6 | 353.12±3.36 | 102.24±.369 |
| | $MPLight^3$ | 350.22±14.0 | 200.44±6.86 | 369.94±12.7 | 151.28±36.4 |
| | $MPLight^{10}$ | 367.37±22.1 | 163.57±66.8 | 365.69±4.49 | 238.07±114 |
| | $MVN^{0.1}$ | 359.25±7.66 | 430.45±11.0 | 403.33±15.8 | 567.84±444 |
| | $MVN^{0.3}$ | 424.05±84.3 | 158.50±36.6 | 407.32±2.00 | 132.47±6.92 |
| | $MVN^1$ | 637.78±31.9 | 243.82±60.2 | 581.95±72.2 | 980.87±260 |
| | $MVN^3$ | 734.60±70.6 | 513.91±56.0 | 742.53±53.4 | 932.41±246 |
| | $MVN^{10}$ | 778.98±68.2 | 540.02±49.6 | 628.43±56.2 | 1021.9±216 |
| Original Reward | CoLight | 333.11±2.89 | 148.41±3.23 | 349.01±.172 | 78.038±14.9 |
| | PressLight | 334.01±1.53 | 160.72±21.5 | 357.83±.981 | 112.29±9.12 |
| | MPLight | 348.02±1.78 | 126.35±14.0 | 353.00±2.27 | 123.20±37.6 |
| | MVN | 436.82±34.5 | 491.23±12.9 | 445.74±6.17 | 173.66±104 |
| | EMVLight | 435.22±7.36 | 130.39±.783 | 385.90±2.78 | 186.31±41.8 |
| | SplitEMV | 333.18±1.12 | 38.161±1.28 | 345.77±.468 | 48.960±3.02 |

vehicles consistently declines—particularly when $\beta > 1$, where the performance drops sharply. However, the efficiency of emergency vehicles does not exhibit proportional improvement and typically peaks at moderate values such as $\beta = 0.1$ or $\beta = 0.3$.

These findings underscore a fundamental trade-off: small $\beta$ values may yield modest gains for emergency vehicles without significantly impairing regular traffic flow, whereas large $\beta$ values induce severe traffic imbalance and increased overall congestion, which ultimately deteriorates both control

Table 8: Performance comparison of different communication strategies.

| Method | SplitEMV | GNN | No Commu. |
|--------|----------|-----|-----------|
| JN-RV | 333.2±1.1 | 346.9±3.3 | 348.0±3.0 |
| JN-EMV | 38.2±1.3 | 41.3±0.5 | 39.5±1.8 |
| HZ-RV | 345.8±0.5 | 347.2±0.3 | 347.5±0.9 |
| HZ-EMV | 49.0±3.0 | 55.9±3.3 | 49.5±7.7 |

Table 9: Performance comparison across different sampling rates.

| Rate | 1/100 | 1/200 | 1/1000 | 1/2000 | 1/5000 |
|------|-------|-------|--------|--------|--------|
| JN-RV | 333.5±1.5 | 332.0±0.6 | 333.2±1.1 | 332.1±0.3 | 331.8±0.6 |
| JN-EMV | 38.9±0.5 | 26.1±0.9 | 38.2±1.3 | 115.1±4.0 | 40.6±0.9 |
| HZ-RV | 346.0±0.8 | 346.0±0.3 | 346.1±0.8 | 346.7±0.4 | 346.3±0.3 |
| HZ-EMV | 21.8±0.1 | 27.2±0.6 | 50.2±4.2 | 34.7±1.6 | 62.3±5.3 |

objectives. This result highlights the limitations of naive reward reweighting for achieving joint optimization in emergency-aware traffic signal control.

## N    EFFECTIVENESS OF COMMUNICATION IN SPLITEMV

We conducted ablation experiments on the communication module, comparing the cases of directly using GNN and not conducting communication at all. The experimental results are shown in Table 8. From the table, we can find that when compared with SplitEMV, whether using GNN to replace the communication module or directly not conducting communication decline the performance.

## O    EFFECT OF THE EMV APPEARING RATE

We analyze the effect of the EMV appearing rate to SplitEMV. Different EMV rate occurs commonly in real situations. If traffic signals are set near key facilities (such as hospitals and police stations), the appearance frequency of EMVs may be relatively high. However, as the shown in Figure 1 (c), ASM is able to construct good strategies with arbitrary EMV appearing rates.

We also tested the model's performance under different appearance frequencies of EMVs, and Table **??** shows the results, which reflects that the superior performance of ASM and SplitEMV is independent of the environment. It should be noted that since the EMV set changes when the EMV appearning frequency changes, **the traffic efficiency of EMVs under different frequencies and datasets cannot be compared with each other**.

## P    TRAINING AND INFERENCE TIME ANALYSIS

The time consumed for algorithm training and inference are shown in the following Table. Given that SplitEMV contains two networks, its running time is indeed slightly longer than that of the existing algorithms. However, its processing speed is still within the same order of magnitude, and compared with the control frequency (10 seconds), this additional time can be almost negligible.

The time consumed for algorithm training and inference are shown in Table 10. Although our training is divided into three stages, the number of training frames is kept the same as other methods. Specifically, all compared methods trained 120,000 frames, while our three stages trained 60,000, 30,000, and 30,000 frames respectively. As for the training time, since EMVLight can make better use of multi-threading, it trains faster than other algorithms, and SplitEMV needs to perform forward propagation for two models in the second half of training, so the time consumption is slightly increased. For inference time, it can be seen that although our method is slightly slower than that of EMVLight, the time difference is not significant, and compared with the control frequency (10 seconds), both are at a negligible level.

Table 10: Training and inference time comparison of different methods.

| Method | SplitEMV | CoLight | PressLight | MPLight | MVN | EMVLight |
|---|---|---|---|---|---|---|
| **Train (min)** | 1148 | 839 | 944 | 911 | 1007 | 593 |
| **Inference (ms)** | 23.6 | 17.5 | 20.3 | 17.9 | 16.4 | 22.6 |

## Q  APPLICATIONS AND LIMITATION

The Decoupled Learning and Adaptive Strategy Merging method we proposed can significantly enhance the efficiency of emergency vehicles without compromising that of regular vehicles, and it is adaptable to diverse traffic densities. Meanwhile, the SplitEMV method achieves performance comparable to the baseline when emergency vehicles are absent and can substantially boost the efficiency of emergency vehicles when they are present. This method has no additional hyper-parameters, eliminating the need for parameter tuning across different scenarios. Furthermore, through the use of Adaptive Strategy Merging, the capacity to improve emergency vehicle efficiency can be integrated with any existing traffic signal control method, thereby enhancing the efficiency of emergency vehicles for existing algorithms. This characteristic significantly broadens the application scenarios of this method, rendering it more suitable for real-world implementations. In addition, our method, aside from its application in solving traffic signal control problems, also holds the potential for application in other multi-objective reinforcement learning problems. Specifically, when multiple objectives are independent of one another, this method can more effectively differentiate tasks without introducing hyper-parameters, facilitating agents in better achieving multiple objectives. However, one shortcoming of this work is that, as this method is founded on the value-based approach, namely learning the state-action function, its application to the policy-based approach proves challenging. Some potential solutions involve adjusting the action probability based on the variance of different action logits or based on the Critic network.

