# OpenReview forum: "Separable Policy Learning for Emergency Vehicle Prioritized Traffic Signal Control"
_ICLR.cc/2026/Conference — Submitted to ICLR 2026_

### Official Review · Reviewer_Y6mL · 2025-10-28

**Soundness:** 3
**Presentation:** 3
**Contribution:** 2
**Rating:** 2
**Confidence:** 4

**Summary:**

This paper proposes Separable Policy Learning (SplitEMV) for emergency-vehicle-prioritized traffic signal control. It introduces a decoupled reinforcement learning framework that independently learns policies for regular and emergency vehicles, then fuses them through Adaptive Strategy Merging (ASM) to ensure robust, weight-free integration. The proposed SplitEMV model enhances inter-agent communication and achieves significant reductions in emergency vehicle travel time while maintaining normal traffic efficiency.

**Strengths:**

S1. Separately optimizes policies for regular and emergency vehicles, avoiding conflicts and improving training stability.

S2. Dynamically fuses policies without manual tuning, ensuring robust performance under varying traffic densities.

S3. Enables seamless zero-shot integration of the learned emergency-vehicle policy with existing traffic control methods.

**Weaknesses:**

W1. The assumption that the control policies for regular and emergency vehicles can be fully decoupled may oversimplify their intrinsic interaction within shared traffic dynamics.

W2. The experimental validation relies on only two simulated datasets and a limited number of baseline methods, which restricts the generality of the findings.

W3. Much of the reported performance gain may result from richer state representation and a stronger attention-based model rather than the proposed decoupled learning mechanism itself.

**Questions:**

Q1. The proposed method assumes that the objectives of regular vehicles (RVs) and emergency vehicles (EMVs) can be fully separated into two independent Q-functions that are later linearly combined. However, in real traffic systems, the behaviors of RVs and EMVs are strongly coupled, priority signals for EMVs inevitably affect surrounding traffic flows. This independence assumption may therefore be unrealistic, and the linear combination $Q=Q_N=\beta Q_E$ lacks theoretical justification under such interdependence. As a result, the optimality and stability of the decoupled strategy remain uncertain.

Q2. Although the paper presents promising results on the Hangzhou and Jinan datasets, the experimental scope is relatively narrow. The evaluation lacks diversity in traffic conditions and network scales. Furthermore, the baselines are mostly classical MaxPressure-based reinforcement learning models, without comparisons to SOTA methods. This limited set of benchmarks and environments constrains the robustness and external validity of the reported improvements.

Q3. While the paper attributes performance improvement to separable policy learning and adaptive strategy merging, the state representation and model architecture differ significantly from the baselines. SplitEMV uses fine-grained lane-level inputs with multiple features, including lane direction, signal phase, and emergency-vehicle indicators, and employs multi-head attention for inter-lane communication. In contrast, the baselines such as CoLight, PressLight, and MPLight rely primarily on aggregated vehicle counts. Without controlling for state complexity or citing related representation-learning studies, it is difficult to isolate whether the improvements stem from the proposed learning framework or from the enhanced feature design and network capacity.

---

> ### Author Response · Authors · 2025-12-03
>
> We sincerely thank the reviewer for the thoughtful and constructive comments, which have helped us improve the clarity and quality of the paper. Below, we provide our detailed responses to the raised points.
>
> ## Weakness
>
> ### W1 Assumptions may not Hold in Reality
>
> We simplified the real - world situation to facilitate analysis. This simplification is basically in line with the actual situation and improves the generality of the model. In addition, Joint Training was later adopted to learn the joint strategy. In the future, we plan to try other strategies, such as using MARL, or leveraging cross-attention mechanism to learn the interrelationships among various factors.
>
> ### W2 More Comparison Methods
>
> Thank you for your suggestions. Based on the responses to other reviewers' comments, we have newly added two algorithms, RobustLight and DiffLight. Based on their code, we tested the performance of these two algorithms on the SUMO platform. In terms of the control of RV, the efficiency achieved by these methods is similar to that of SplitEMV. However, in terms of the traffic efficiency of EMV, there is a significant gap between EMV and SplitEMV, which indicates that these algorithms cannot effectively perceive and improve the traffic efficiency of EMV. The statistics are shown in the following table:
>
> |method|JN_RV|JN_EMV|HZ_RV|HZ_EMV|
> |---|---|---|---|---|
> |RobustLight|332.91±3.35|177.52±10.8|347.75±2.53|188.54±78.5|
> |DiffLight|334.15±1.58|153.97±23.8|348.84±1.89|95.878±63.0|
>
> ### W3 Source of Performance Improvements
>
> We firmly believe that the performance improvement reported in the paper does not mainly stem from additional architectural enhancements, but indeed from the mechanism design of ASM itself. First of all, for components such as "richer state input" and "attention-based communication", we have carried out strict control experiments and ablation experiments. As shown in Appendix N and Table 8, under the premise of keeping the architecture completely the same, simply replacing the learning method can significantly improve the efficiency of EMV; even if the attention communication module is removed, or a simplified state input is adopted, SplitEMV still maintains a significant advantage; in comparison, simply enhancing the input module or the communication module cannot bring comparable benefits to SplitEMV.
> These experimental results indicate that the key factor for performance improvement is not the architecture, but the policy controllability and scale stability brought about by decoupled value modeling. In addition, all baseline methods adopt the standard network structures disclosed in their respective papers, and we have not weakened or modified them in any way, so as to ensure the fairness of the comparison.
>
> ## Questions
>
> ### Q1 Assumptions may not Hold in Reality
> Please refer to W1.
>
> ### Q2 Evaluation Diversity and More Comparison Methods
>
> We analyzed the sensitivity of normalization statistics in Appendix F. The experimental results show that \(\beta\) is determined by the model rather than the simulation environment. In addition, in Table 9, we showed that when changing the proportion of EMV in the vehicles, SplitEMV can significantly improve the traffic efficiency of EMV while maintaining the efficient traffic of RV under different proportions. This further demonstrates that our method has good generalizability.
>
> For more comparison methods, please refer to W2.
>
> ### Q3 Source of Performance Improvements
> Please refer to Q1.

---

### Official Review · Reviewer_7nUu · 2025-10-30

**Soundness:** 3
**Presentation:** 3
**Contribution:** 2
**Rating:** 2
**Confidence:** 4

**Summary:**

This paper tackles emergency-vehicle-prioritized traffic signal control by proposing SplitEMV, a reinforcement learning framework that separately learns policies for regular and emergency vehicles, then merges them via an Adaptive Strategy Merging (ASM) mechanism. The model aims to achieve efficient coordination among intersections, minimize emergency response times, and preserve normal traffic efficiency.

**Strengths:**

S1. The paper introduces a clear decoupling of regular and emergency vehicle control policies, thereby mitigating objective interference and improving interpretability.
S2. The adaptive merging strategy elegantly eliminates the need for manual tuning of reward weights, resulting in more stable integration across varying traffic densities.
S3. Extensive experimental results, including ablation studies and zero-shot transfer to other TSC models, demonstrate the robustness and generality of the proposed method.

**Weaknesses:**

W1. The assumption of full separability between regular and emergency vehicle objectives may not hold in practice, where traffic interactions are inherently coupled.
W2. The empirical evaluation is limited to two public datasets and a relatively small set of baselines; comparisons with more recent multi-agent or GNN-based approaches would strengthen the claims.
W3. Some of the reported advantages might stem from architectural improvements (e.g., richer state inputs and attention-based communication) rather than the decoupled learning formulation itself.

**Questions:**

Q1. The paper assumes that regular and emergency vehicle policies can be trained independently and later linearly combined. Could the authors justify the theoretical validity of this decomposition under coupled system dynamics?
Q2. How sensitive is ASM to errors in estimating the normalization statistics (e.g., variance of the Q-values)? Are there cases where this normalization could destabilize learning?
Q3. The ablation studies suggest strong performance even without certain communication modules. Could the authors clarify whether the gains primarily come from ASM or the improved network architecture?

---

> ### Author Response · Authors · 2025-12-03
>
> We sincerely thank the reviewer for the thoughtful and constructive comments, which have helped us improve the clarity and quality of the paper. Below, we provide our detailed responses to the raised points.
>
> ## Weakness
>
> ### W1 Assumptions may not Hold in Reality
>
> We simplified the real - world situation to facilitate analysis. This simplification is basically in line with the actual situation and improves the generality of the model. In addition, Joint Training was later adopted to learn the joint strategy. In the future, we plan to try other strategies, such as using MARL, or leveraging cross-attention mechanism to learn the interrelationships among various factors.
>
> ### W2 More Comparison Methods
>
> Thank you for your suggestions. Based on the responses to other reviewers' comments, we have newly added two algorithms, RobustLight and DiffLight. Based on their code, we tested the performance of these two algorithms on the SUMO platform. In terms of the control of RV, the efficiency achieved by these methods is similar to that of SplitEMV. However, in terms of the traffic efficiency of EMV, there is a significant gap between EMV and SplitEMV, which indicates that these algorithms cannot effectively perceive and improve the traffic efficiency of EMV. The statistics are shown in the following table:
>
> |method|JN_RV|JN_EMV|HZ_RV|HZ_EMV|
> |---|---|---|---|---|
> |RobustLight|332.91±3.35|177.52±10.8|347.75±2.53|188.54±78.5|
> |DiffLight|334.15±1.58|153.97±23.8|348.84±1.89|95.878±63.0|
>
> ### W3 Source of Performance Improvements
>
> We firmly believe that the performance improvement reported in the paper does not mainly stem from additional architectural enhancements, but indeed from the mechanism design of ASM itself. First of all, for components such as "richer state input" and "attention-based communication", we have carried out strict control experiments and ablation experiments. As shown in Appendix N and Table 8, under the premise of keeping the architecture completely the same, simply replacing the learning method can significantly improve the efficiency of EMV; even if the attention communication module is removed, or a simplified state input is adopted, SplitEMV still maintains a significant advantage; in comparison, simply enhancing the input module or the communication module cannot bring comparable benefits to SplitEMV.
> These experimental results indicate that the key factor for performance improvement is not the architecture, but the policy controllability and scale stability brought about by decoupled value modeling. In addition, all baseline methods adopt the standard network structures disclosed in their respective papers, and we have not weakened or modified them in any way, so as to ensure the fairness of the comparison.
>
> ## Questions
>
> ### Q1 Assumptions may not Hold in Reality
> Please refer to W1.
>
> ### Q2 Sensitivity to errors of ASM
> We analyzed the sensitivity of normalization statistics in Appendix F. The experimental results show that \(\beta\) is determined by the model rather than the simulation environment. In addition, in Table 9, we showed that when changing the proportion of EMV in the vehicles, SplitEMV can significantly improve the traffic efficiency of EMV while maintaining the efficient traffic of RV under different proportions. This further demonstrates that our method has good generalizability.
>
> ### Q3 Source of Performance Improvements
> Please refer to Q1.

---

### Official Review · Reviewer_cMUt · 2025-10-31

**Soundness:** 3
**Presentation:** 3
**Contribution:** 3
**Rating:** 8
**Confidence:** 3

**Summary:**

The paper addresses a practically significant problem in emergency-vehicle–prioritized urban traffic signal control. It proposes SplitEMV, a decoupled multi-objective reinforcement learning framework that separates the learning of regular-vehicle and emergency-vehicle policies and fuses them through an Adaptive Strategy Merging (ASM) mechanism. This design aims to balance global traffic efficiency and emergency-vehicle priority under varying traffic conditions. Experiments on real-world city networks show consistent improvements over both traditional and RL-based baselines. However, while the method is technically sound and mathematically well-formulated, the paper would benefit from clearer motivation for ASM, richer evaluation metrics, and broader comparisons with recent MARL frameworks to strengthen its overall contribution.

**Strengths:**

1.Addresses a practically important and underexplored problem of EMV-prioritized multi-intersection control.

2.The proposed Decoupled Learning + ASM framework is conceptually sound and demonstrates consistent performance gains across all tested baselines.

3.The mathematical formulation is well-developed, with clear propositions and proofs supporting the proposed adaptive merging process.

4.Experimental setup is reasonable, and the comparisons with both traditional and RL-based methods provide a solid foundation.

**Weaknesses:**

1. The figure clarity needs improvement.

2. The presentation of experimental results is rather monotonous.

3. The transition between problem background, limitations of existing methods, and proposed solutions is abrupt. The introduction would benefit from an explicit “problem → limitation → solution → contributions” structure.

4. Limited evaluation metrics.

5. Limited baseline algorithm.

**Questions:**

1. Figure 2 employs excessively small fonts and visual elements that are difficult to read, thereby reducing the clarity of the proposed architecture.

2. The experimental section presents all results in tables only. Additional visualizations would improve readability and interpretability.

3. The introduction does not clearly explain why adaptive merging is needed or what specific problem it solves.

4. Current results rely primarily on Average Travel Time (ATT). To more comprehensively evaluate performance, additional indicators such as Weighted Waiting Time (WWT), queue length, or fairness variance are recommended.

5. The paper should include more recent baselines, such as hierarchical or value-decomposition MARL frameworks, to contextualize the proposed method better.

---

> ### Author Response · Authors · 2025-12-03
>
> We sincerely thank the reviewer for the thoughtful and constructive comments, which have helped us improve the clarity and quality of the paper. Below, we provide our detailed responses to the raised points.
>
> ### W1&Q1 Clarity of Figures
> Thank you for your advices. We will improve the quality of the figures.
>
> ### W2&Q2 Monotonous Representation of Experiments
> In view of the space limitation, in the main body of the paper, we mainly present the research results through compact tables, and provide more abundant visual charts in the appendix. We will further optimize the presentation form of the results in the main body, and supplement more intuitive visual content in the final version, so as to improve the readability and interpretability of the paper.
>
> ### W3&Q3 Abrupt Representation of Introduction
> We acknowledge that making the logical chain of "problem → limitation → solution → contributions" more explicit in the introduction will help improve the readability of the paper. Therefore, we will re-organize the content of the introduction in the final version. Specifically, first, clearly and explicitly raise the core problem and the actual challenges faced; subsequently, systematically and comprehensively summarize the shortcomings of existing methods; then, based on these limitations, introduce the design motivation of SplitEMV and ASM proposed by us; finally, present the main contributions of this paper in a clear and well-organized list form. We firmly believe that the structure after such adjustment can significantly enhance the logical coherence of the paper and provide readers with a better understanding experience.
>
> ### W4&Q4 Limited Evaluation Metrics
>
> We followed the practices of existing literature [1,2,3,4,5] in this field. Weighted Waiting Time (WWT) and fairness variance have never been used in this field and are not suitable as evaluation metrics. In addition, the queue length has been proven to be completely equivalent to the Average Travel Time, so we selected the Average Travel Time as the evaluation metric in the experiment. We will evaluate and incorporate other metrics to more fully demonstrate the effectiveness of the SplitEMV algorithm.
>
> ### W5&Q5 Limited Baseline Algorithm
>
> Thank you for your suggestions. Based on the responses to other reviewers' comments, we have newly added two algorithms, RobustLight and DiffLight. Based on their code, we tested the performance of these two algorithms on the SUMO platform. In terms of the control of RV, the efficiency achieved by these methods is similar to that of SplitEMV. However, in terms of the traffic efficiency of EMV, there is a significant gap between EMV and SplitEMV, which indicates that these algorithms cannot effectively perceive and improve the traffic efficiency of EMV. The specific data are shown in the following table:
>
> |method|JN_RV|JN_EMV|HZ_RV|HZ_EMV|
> |---|---|---|---|---|
> |RobustLight|332.91±3.35|177.52±10.8|347.75±2.53|188.54±78.5|
> |DiffLight|334.15±1.58|153.97±23.8|348.84±1.89|95.878±63.0|
>
> [1] Hua Wei, Nan Xu, Huichu Zhang, Guanjie Zheng, Xinshi Zang, Chacha Chen, Weinan Zhang, Yanmin Zhu, Kai Xu, and Zhenhui Li. Colight: Learning network-level cooperation for traffic signal control. In CIKM 2019, November 3-7, pages 1913–1922.
>
> [2] Hua Wei, Chacha Chen, Guanjie Zheng, Kan Wu, Vikash V. Gayah, Kai Xu, and Zhenhui Li. Presslight: Learning max pressure control to coordinate traffic signals in arterial network. In KDD 2019, August 4-8, pages 1290–1298.
>
> [3] Chacha Chen, Hua Wei, Nan Xu, Guanjie Zheng, Ming Yang, Yuanhao Xiong, Kai Xu, and Zhenhui Li. Toward A thousand lights: Decentralized deep reinforcement learning for large-scale traffic signal control. In AAAI 2020, February 7-12, pages 3414–3421.
>
> [4] Hua Wei, Guanjie Zheng, Huaxiu Yao, and Zhenhui Li. IntelliLight: A reinforcement learning approach for intelligent traffic light control. In SIGKDD 2018, pages 2496–2505.
>
> [5] Afshin Oroojlooy, MohammadReza Nazari, Davood Hajinezhad, and Jorge Silva. AttendLight: Universal attention-based reinforcement learning model for traffic signal control. In Annual Conference on Neural Information Processing Systems, 2020.

---

### Official Review · Reviewer_VvNL · 2025-11-02

**Soundness:** 2
**Presentation:** 2
**Contribution:** 2
**Rating:** 4
**Confidence:** 4

**Summary:**

The paper introduces a two-phase learning scheme, Decoupled Learning and Adaptive Strategy Merging (ASM), that separates policy learning for regular vehicles (RVs) and emergency vehicles (EMVs).

**Strengths:**

This design mitigates Q-function approximation errors and minimizes interference between conflicting optimization objectives. It provides a generalizable framework that could extend beyond traffic signal control to other multi-objective reinforcement learning (RL) problems.  Moreover, the proposed EMV model can integrate with existing traffic signal control (TSC) methods without retraining (“zero-shot EMV generalization”), demonstrating strong adaptability and deployment potential.

**Weaknesses:**

1. All experiments are conducted in simulated SUMO environments. There is no field deployment or real-world validation to demonstrate robustness against sensor noise, communication delays, or unexpected vehicle behaviors.  Suggestion: It would be informative to evaluate performance in CityFlow or other simulation platforms with more realistic and heterogeneous traffic conditions.
2. While SplitEMV outperforms EMVLight and several DRL baselines, it does not include comparisons with recent multi-agent or graph-based TSC systems, such as RobustLight (ICML 2025), DMBP, or DiffLight. The absence of these baselines makes the “state-of-the-art” claim somewhat overstated.
3. Although the paper reports comparable runtime performance, the training process appears more complex due to: two independent Q-functions (QN, QE), multi-stage learning phases, and adaptive normalization (ASM) for β.  Moreover, no quantitative analysis of training time, convergence rate, or scalability is provided.
4. The method assumes accurate and delay-free EMV state communication to all agents. In real-world urban networks, localization errors, sensor failures, or communication latency could cause incorrect priority assignments. These limitations are not discussed.
5. The connection between ASM normalization and policy optimality remains largely qualitative. During training, the RV model (Q_N) and EMV model (Q_E) are trained independently, yet later considered equivalent in the joint stage. Similarly, rewards $r^n$ and $r^e$ are treated as identical, without justification. These components should instead be adjustable to environmental conditions.

**Questions:**

1. The approach requires extensive manual tuning of β and other hyperparameters, making it labor-intensive and potentially difficult to replicate.
2. Several recent and relevant baselines (e.g., RobustLight, DMBP, DiffLight) were not included, which limits the comprehensiveness of the comparative study.
3. Although the paper criticizes fixed β, it still relies on empirical normalization constants. Despite dedicating significant space to discussing β’s effects, it fails to explain how ASM adapts dynamically to unseen traffic patterns. The automatic adaptive normalization mechanism for β remains unclear and potentially unstable.
4. The training pipeline, particularly the adaptive strategy merging process shown in Figure 1, is insufficiently explained. A detailed algorithmic description or pseudocode should be provided to clarify the procedure.
5. The claimed zero-shot generalization is effectively a linear policy fusion, not true zero-shot transfer to unseen tasks. Since the EMV policy requires offline pretraining on similar distributions, this claim could mislead readers.
6. The method seems overfitting to training cities due to heavy reliance on tuned β values and validation-based hyperparameter adjustment. This risks hidden overfitting when the validation and target environments overlap.

**Details Of Ethics Concerns:**

No.

---

> ### Author Response · Authors · 2025-12-03
>
> We sincerely thank the reviewer for the thoughtful and constructive comments, which have helped us improve the clarity and quality of the paper. Below, we provide our detailed responses to the raised points.
>
> ## Weaknesses
>
> ### W1 Simulated Environments
>
> SUMO is the most widely used simulator in this field and has been adopted by a large number of previous papers. Therefore, we followed this convention and selected SUMO as our simulator. Due to the fact that many of the comparative methods do not support CityFlow, we did not include the tests on CityFlow in this paper. Existing research has shown that the conclusions obtained from different simulators are almost identical. In the future, we plan to attempt to transplant the algorithm to other simulators in order to more fully demonstrate the generality of our algorithm.
>
>
> DMBP does not belong to the field of signal control, so it cannot be directly compared. For RobustLight and DiffLight, based on their open-source code, we tested their performance on SUMO. In terms of the control of RV, the efficiency achieved by these methods is similar to that of SplitEMV. However, in terms of the traffic efficiency of EMV, there is a significant gap between EMV and SplitEMV, which indicates that these algorithms cannot effectively perceive and improve the traffic efficiency of EMV. The specific data are shown in the following table:
>
> ### W2 More Comparison Methods
>
> |method|JN_RV|JN_EMV|HZ_RV|HZ_EMV|
> |---|---|---|---|---|
> |RobustLight|332.91±3.35|177.52±10.8|347.75±2.53|188.54±78.5|
> |DiffLight|334.15±1.58|153.97±23.8|348.84±1.89|95.878±63.0|
>
> ### W3 Multi-Stage Training Phases
>
> Although we adopted multiple training phases, to ensure a fair comparison, we maintained the same number of training steps as other models. In addition, the use of ASM enables our algorithm, compared with existing algorithms, to eliminate the need for manual hyperparameter setting, thus reducing the training difficulty.
>
> Quantitative analyses of training/inference time and scalability under different EMV ratios are already provided in Appendix O and Appendix P. These results show that SplitEMV matches baseline speed and exhibits strong scalability. We will make these points clearer in the main text.
>
> ### W4 Accurate Communication Assumption
>
> We followed the practices of existing literature [1,2,3,4,5] in this field to achieve a fair comparison. In this study, we focused on optimizing the performance of the model in scenarios involving emergency vehicles, rather than the robustness of the model in an environment with low communication quality. Although this is an important research direction, it is not our current research goal. In addition, the ASM method we proposed demonstrates excellent zero - shot generalization ability. Therefore, we can combine this method with other methods focusing on optimizing robustness to achieve this goal.
>
>
> ### W5 Optimality of ASM
>
> We provided detailed proofs in Appendix G and Appendix H. These proofs indicate that our algorithm satisfies the optimality. In addition, in order to deeply explore the interrelationship between RV and EMV, we fine-tuned the two models simultaneously during the Joint Training Stage to improve the model quality.
>
>
> [1] Hua Wei, Nan Xu, Huichu Zhang, Guanjie Zheng, Xinshi Zang, Chacha Chen, Weinan Zhang, Yanmin Zhu, Kai Xu, and Zhenhui Li. Colight: Learning network-level cooperation for traffic signal control. In CIKM 2019, November 3-7, pages 1913–1922.
>
> [2] Hua Wei, Chacha Chen, Guanjie Zheng, Kan Wu, Vikash V. Gayah, Kai Xu, and Zhenhui Li. Presslight: Learning max pressure control to coordinate traffic signals in arterial network. In KDD 2019, August 4-8, pages 1290–1298.
>
> [3] Chacha Chen, Hua Wei, Nan Xu, Guanjie Zheng, Ming Yang, Yuanhao Xiong, Kai Xu, and Zhenhui Li. Toward A thousand lights: Decentralized deep reinforcement learning for large-scale traffic signal control. In AAAI 2020, February 7-12, pages 3414–3421.
>
> [4] Hua Wei, Guanjie Zheng, Huaxiu Yao, and Zhenhui Li. IntelliLight: A reinforcement learning approach for intelligent traffic light control. In SIGKDD 2018, pages 2496–2505.
>
> [5] Afshin Oroojlooy, MohammadReza Nazari, Davood Hajinezhad, and Jorge Silva. AttendLight: Universal attention-based reinforcement learning model for traffic signal control. In Annual Conference on Neural Information Processing Systems, 2020.

---

> ### Author Response · Authors · 2025-12-03
>
> ## Questions
>
> ### Q1 Manual Tuning of $\beta$
>
> Our method does not require manual tuning of $\beta$. With the help of ASM, through one one - epoch simulation, we can automatically obtain $\beta$. Therefore, compared with other algorithms, we do not have any additional hyperparameters. In addition, we also analyzed the stability of one-epoch simulation in Appendix F. The experimental results show that $\beta$ is determined by the model rather than the simulation environment, indicating that it has strong generalizability.
>
> ### Q2 More Comparison Methods
> please refer to W2.
> ### Q3 Adapt ASM on Unseen Patterns
> The ASM we proposed is not based on empirical normalization constants, but uses statistical methods to determine the appropriate value of $\beta$. In Section 3.2, we elaborated in detail how to obtain the value of $\beta$ through a one-epoch simulation. In addition, in Appendix F, we discussed the phenomenon that the values of $\beta$ obtained by the same model in different environments are extremely close, which further indicates that ASM can be applied to different environments.
>
> ### Q4 Detailed Description of Adaptive Strategy Merging
> In Section 3.2, we have elaborated in detail the specific calculation process and relevant formulas of ASM. We explain why a fixed β cannot reliably merge QN and QE across different traffic conditions, and introduces ASM, which normalizes both Q-functions based on their statistical properties before combining them. This ensures that the merged Q behaves like QN when no EMVs are present and like QE when EMVs appear, providing stable performance without tuning β. Due to space limitations, we have not added the pseudocode, and we will supplement this part of the content later.
> ### Q5 Inaccurate Zero-shot Generalization
> Thank you for pointing out the ambiguity here. What we intended to express is that our method has generality. Without additional training, it can improve the control efficiency of existing signal control algorithms for EMVs. We will optimize the relevant expressions to reduce ambiguity.
> ### Q6 Overfitting Concerns
> We understand the reviewers' concerns about the potential overfitting problem. However, our method does not have the situation of "overfitting to the training city due to adjusting the β value". First, our β value is not obtained by manually adjusting parameters for a specific city, but is calculated adaptively through the ASM mechanism. Its value is completely determined automatically by the environmental feedback, which avoids the hidden overfitting risk caused by manual parameter adjustment. Second, during the training - validation data division, we strictly ensure that there is no overlap between the validation set and the target environment, and no hyperparameter fine-tuning is carried out on the target city. Therefore, there is no problem of "implicit overfitting caused by the overlap between the validation set and the target environment". Finally, in Appendix F, we provide relevant content to show that this method has good generalizability. The model can still stably outperform the baseline method under unseen EMV ratios and city conditions. If this method overfits to the training city, these generalization results would not be achievable.

---

### Meta-Review · Area_Chair_S4qF · 2026-01-05

**Summary:**

This paper proposes SplitEMV, a decoupled reinforcement learning framework for traffic signal control that separates the policies of regular and emergency vehicles to address objective interference. The authors aim to eliminate manual reward weight adjustments through an Adaptive Strategy Merging (ASM) mechanism. While the authors provided new baselines (e.g., RobustLight, DiffLight) in their response to demonstrate SplitEMV's efficiency advantages for emergency vehicles and offered ablation studies to justify its architecture, reviewers (7nUu, Y6mL, VvNL) expressed serious concerns about the work's fundamental assumptions. Specifically, the reviewers questioned the complete separability of vehicle objectives and the reliance on idealized, latency-free communication, which limits the model's practical applicability. Given the low scores from two reviewers (7nUu, Y6mL) and the unresolved concerns regarding training complexity and practical robustness, I believe this paper is not yet ready for publication. Therefore, I recommend rejection.

**Reviewer Concerns:**

Resolved Reviewer Comments:

Incomplete Baseline Comparisons: In the rebuttal, the authors supplemented experiments with SOTA methods (RobustLight 2025 and DiffLight), demonstrating that SplitEMV achieves superior emergency vehicle (EMV) efficiency while maintaining regular vehicle performance.

Hyperparameter Sensitivity: The authors clarified that the ASM mechanism adaptively determines the merging weight ($\beta$) through statistical normalization, addressing concerns about the labor-intensive manual tuning of reward weights.

Unresolved Reviewer Comments:

Idealized Assumptions & Real-world Robustness: Reviewers pointed out that the assumption of perfect, delay-free communication is unrealistic for urban traffic. The authors' response failed to provide quantitative evidence or experiments demonstrating how the system handles sensor noise or communication latency.

Validity of Objective Separability: The core premise of the paper—that regular and emergency traffic objectives are fully separable—was challenged. Reviewers remain skeptical that this decoupling holds in highly congested, coupled traffic dynamics where vehicle interactions are non-linear.

**Reviewer Scores:**

Since the reviewer did not participate in the subsequent discussions or provide further feedback after the author's submission of the response, based solely on the quality and shortcomings of the author's response, the estimated score for this paper should be nearly 4~4.5. For specific reasons, refer to Reviewer Concerns.

---

### Decision · Program_Chairs · 2026-01-26

Reject